



# Spy4Cast v1.0: a Python Tool for statistical seasonal forecast based on Maximum Covariance Analysis

Pablo Duran-Fonseca[1] and Belén Rodríguez-Fonseca[1,2]

[1]Universidad Complutense de Madrid, Madrid, Spain
[2]Instituto de Geociencias IGEO, CSIC-UCM, Madrid, Spain

**Correspondence:** Pablo Duran-Fonseca (pdrm56@gmail.com). Belén Rodríguez-Fonseca (brfonsec@ucm.es).

**Abstract.** Maximum Covariance Analysis (MCA) is a well known discriminant analysis technique used for finding coupled patterns in climate data. This is a powerful tool that has been applied to the study of teleconnections, by reducing all possible relationships between a predictor and a predictand field to a few modes of covariability patterns. MCA can be used to provide statistical forecasts, which can complement predictions performed with dynamical models. Nevertheless, the power of this tool relies on its application in a productive and easy way, as it can be applied to the huge climate data-sets available. Spy4Cast is an open-source interface (API), implemented in Python, that contains a MCA-based statistical model to be used for seasonal forecast. Its main goal is to increase automation and productivity. Spy4Cast enables large data-set manipulation and also performs basic tasks like region slicing and plotting. The methodology consists on an initial configuration (data-set reading and slicing) and preprocessing that prepares the data to be fed into MCA, crossvalidation and validation. It acts upon any kind of predictor and predicting variables that can come from any source of data. Spy4Cast analyses the model sensitivity to particular years, including a diagnosis of the stability of the obtained modes to particular outliers. Finally, the spatial and temporal skill, in terms of anomaly correlation coefficient is obtained and a hindcast is provided. The software is easily accessible through a python package and well documented for beginners and experienced programmers. Only a reduced number of third-party libraries are needed, and they are those widely used in data-science and physics.

## 1 Introduction

Maximum Covariance Analysis (MCA) is a well known discriminant analysis technique used for finding coupled patterns in climate data (Bretherton et al., 1992; Liu, 2003). This technique consists on finding linear combinations of the time series of a predictand and a predictor field in a way that their covariance is maximized. This is done through the diagonalization of the covariance matrix, a problem which is solved by using singular value decomposition (SVD). MCA is a powerful tool when wrapping large amount of climate information in a few modes of variability. In this context, MCA analysis provides spatial patterns of the predictor and predictand field which are related by teleconnections. For example, when using as predictor field the tropical Pacific sea surface temperature (SST) anomalies, and, as predictand field, the overlying anomalous sea level pressure (SLP), the leading mode is found to be a couple of anomalous SST and SLP patterns which are highly connected. The SST pattern will have the structure of El Niño and the SLP pattern will be the Southern Oscillation, which is the covarying





SLP pattern forced by El Niño. The time expansion coefficients associated to each mode will represent how strongly each mode loads on each year and will be highly related to the Southern Oscillation and Niño indices. Using the resultant expansion coefficients and singular vectors, MCA can be applied to produce reliable statistical predictions. In general, empirical statistical forecasts can serve both as a baseline for dynamical models and can be used to improve the forecasts by limiting the effects of dynamical model biases (Eden et al., 2015). Other machine learning techniques, involving the use of more complex non-

linear algorithms, can be used for statistical forecast. In this way, neural networks are currently providing accurate predictions (Mekanik et al., 2013; Bi et al., 2023). However, it is still difficult to find the regions in which these deep learning techniques find the predictability and to identify physical relationships between the predictor and the predictand field.

Although MCA asummes linearity when finding coupled patterns, it has the advantage of being easily interpretable, as it provides this base of coupled patterns in which the main relationships can be identified. This methodology has been widely used

in climate variability studies (Bretherton et al., 1992) and it is still used, (Guan et al., 2023). For example, it has been recently used to correct the stratospheric circulation forecast based on the covariability modes between SST and stratospheric variables (Rao et al., 2023). It has multiple applications in climate services as the identification of the climate modes influencing global crop yields (Anderson et al., 2019). Sea surface temperature (SST) is a widely used variable in seasonal prediction analysis, as it is a proxy of the surface heat storage in the ocean mixed layer. Thus, using the sea surface temperature as predictor with

some months in advance, MCA can be used for seasonal predictions (Wilks, 2008; Rodríguez-Fonseca et al., 2006, 2016; Wang et al., 1999). In the last decades, a new paradigm of research in climate variability studies has emerged in relation to the prediction of El Niño from other tropical basins (Cai et al., 2019). In this way, it has been found how, from the 1970's boreal winter El Niño can be predicted from the previous boreal summer Atlantic zonal mode (Rodríguez-Fonseca et al., 2009; Ding et al., 2012). Nevertheless, current coupled models are able to reproduce this relation only using the initialization that contains

the Atlantic Niño, due to the fact that this mode of tropical Atlantic variability is not well reproduced by models (Exarchou et al., 2021). Nevertheless empirical forecasts using MCA provide a realiable alternative (Martín-Rey et al., 2015) to current coupled models predictions due to bias in the representation of the predictors (counillon).

A previous implementation of this methodology that uses SST as predictor, named S4CAST (from "SST-based Statistical Seasonal ForeCAST"), is available for MATLAB (Suárez-Moreno and Rodríguez-Fonseca, 2015). Also, the International Re-

search Institute for Climate and Society (IRI) provides an statistical forecasts tool, name CPT (from "Climate Predictability Tool"), which is a software package for constructing a seasonal climate forecast model, performing model validation, and producing forecasts given updated data using mainly Canonical Correlation Analysis (Mason et al., 2019).

In addition to classical MCA, S4CAST identifies periods with stationary relations between the predictor and the predictand field. This model has been widely used to predict rainfall, extreme events and malaria (Diouf et al., 2022; Diakhaté et al.,

2020; Suárez-Moreno et al., 2018). The present model, named Spy4Cast, is a python library designed to be used as a library to work with large databases. It differs from S4CAST because it is not designed to assess stationarity but to provide , in addition to crossvalidated hindcasts, also predictions using a test period and validation period. The advantages of this library is that, for example, it could be used in the framework of *esmvaltool*, a community diagnostic and performance metrics tool for the evaluation of Earth system models Righi et al. (2020).





Modeling tools used in science are more often written in low-level compiled languages (C, C++, ...) that are usually faster than high-level interpreted languages like Python. However, low-level languages are more difficult to use and require a deeper understanding of software engineering. Code written in C, C++ or FORTRAN is also more difficult to share across different platforms and, as opposed to Python, often require a larger code base. Python is able to maintain performance while being easy, brief and compatible thanks to many open-source libraries which write their core functionality in C or C++ (numpy,

scipy, xarray, ...). This is the reason why this kind of tools have been shifting towards Python in recent years. Spy4Cast takes advantage of these packages and wraps them in a way that the user only has to care about calculations. Spy4Cast is integrated within the newest Python tendency towards static typing, with tools such as mypy, by using type annotations introduced in PEP 526 (Gonzalez et al., 2016). This allows for better code readability, better IDE functionality, easier bug-fixing and, most importantly, higher scalability. It is also tested with a suite of unittests to ensure that it can be improved through out the years

in a safe way.

In the present paper Spy4Cast package is described through its application to a particular database, in order to illustrate an example. The source code for the API is available in GitHub (Duran-Fonseca, 2023) (development version) and in the (Duran-Fonseca and Rodriguez-Fonseca, 2024b) (archived version 1.0.0 used for this paper). The code present in this paper together with the data and examples is in a zenodo folder (Duran-Fonseca and Rodriguez-Fonseca, 2024a). The remainder of

75 the paper is structured as follows. Section 2 describes the Methodology and Data used for implementing using the API. The model structure is explained in section 3. Finally, section 4 shows an example for the prediction of Sahelian rainfall using SSTs from the Pacific.

## 2 Methodology and Data

Spy4Cast implements the methodology of Maximum Covariance (MCA) analysis (Bretherton et al., 1992) which needs a
80 predictor field, $Y$, and a predictand field, $Z$ . These fields are organized in space-time matrices where each row represents the evolution of a certain point in space ($n_y$ and $n_z$ points for $Y$ and $Z$ respectively) across $n_t$ discrete time intervals. When covarying on time, $Y$ and $Z$ must have the same time dimension ($n_t$), even though they can have a certain lag. However, they do not need to have (in general) the same space dimension and so the covariance matrix, defined as $C = Y \cdot Z^T$, has dimensions $(n_y, n_z)$.

Thus, $C$ compresses all possible maps that can be obtained when covarying all time series of all the spatial points in the $Y$ and $Z$ fields. For example, each column in $C$ has $n_y$ dimension, which corresponds to a map that is indicating how all the points in $Y$ covariate with a point in $Z$ . Thus, $C$ holds all $n_y$ $Z-maps$ containing the relation between each point in $Y$ with $Z$. Also, it contains all $n_z$ $Y-maps$ containing the relation between each point in $Z$ with $Y$. The information of this matrix is redundant , as closer points produce the same maps in $C$. Also it is a complex matrix as it takes into account all

possible relations between points. Thus, MCA reduces the dimension of $C$ in a $n_m$ number of covariability modes explaining the maximum amount of covariance, by diagonalizing it (Bretherton et al., 1992).





As $\boldsymbol{C}$ can be non-square, the singular value decomposition (SVD) methodology is applied for diagonalizing it. The resultant outputs are a set $\boldsymbol{n}_m$ singular vectors $\boldsymbol{r}(n_y, n_m)$ and $\boldsymbol{q}(n_m, n_z)$, being $n_m$ the number of modes chosen. The other output is the diagonal matrix of eigenvalues, which represents the squared covariance fraction (scf) explained by each mode. The singular vectors $\boldsymbol{R}_i$ (column) and $\boldsymbol{Q}^i$ (row) for each mode $i$, can be used to define expansion coefficients $\boldsymbol{U}^i$ and $\boldsymbol{V}^i$ which represent the time series row vectors whose covariance is maximized[1]. Thus,

$$\boldsymbol{U} = \boldsymbol{R}^T \cdot \boldsymbol{Y}, \quad \boldsymbol{V} = \boldsymbol{Q} \cdot \boldsymbol{Z}$$

Thus $\boldsymbol{U}$ and $\boldsymbol{V}$ have maximum sample covariance, which can be expressed as:

$$\boldsymbol{C}_{UV} = \boldsymbol{U} \cdot \boldsymbol{V}^T = \boldsymbol{R}^T \cdot \boldsymbol{C} \cdot \boldsymbol{Q}^T$$

Each mode explains a fraction of variance and it is composed by a couple of $\boldsymbol{R}_i$ and $\boldsymbol{Q}^i$ patterns for a particular $i$ mode and a couple of time series $U_i$ and $V_i$ which are linked by having maximum covariance. The classical way to represent the results is by standardizing the expansion coefficients, so they are usually stored as:

$$\boldsymbol{U}_s^i = \frac{\boldsymbol{U}^i - \mathrm{mean}(\boldsymbol{U}^i)}{\mathrm{std}(\boldsymbol{U}^i)}, \quad \boldsymbol{V}_s^i = \frac{\boldsymbol{V}^i - \mathrm{mean}(\boldsymbol{V}^i)}{\mathrm{std}(\boldsymbol{V}^i)}$$

The spatial coupled modes of variability $R_i$ and $Q^i$, are represented as regression and correlation maps ($\boldsymbol{S}_{UY}$ - $\boldsymbol{R}_{UY}$ and $\boldsymbol{S}_{UZ}$ - $\boldsymbol{R}_{UZ}$ respectively). These maps are defined as homogeneous maps for the predictor field and as heterogeneous maps for the predictand field, and can be calculated as follows:

$$\boldsymbol{S}_{UY} = \frac{1}{n_t} \boldsymbol{Y} \cdot \boldsymbol{U}_s^T, \quad \boldsymbol{R}_{UY} = \mathrm{corr}(\boldsymbol{U}_s, \boldsymbol{Y}), \quad \boldsymbol{S}_{UZ} = \frac{1}{n_t} \boldsymbol{Z} \cdot \boldsymbol{U}_s^T, \quad \boldsymbol{R}_{UZ} = \mathrm{corr}(\boldsymbol{U}_s, \boldsymbol{Z}),$$

being corr the correlation operator.

Using the information from the singular vectors and expansion coefficients, a linear regression model of $\boldsymbol{Z}$ ($\hat{\boldsymbol{Z}}$ or predicted $\boldsymbol{Z}$) can be developed (Suárez-Moreno and Rodríguez-Fonseca, 2015), in a way that:

$$\hat{\boldsymbol{Z}} = \boldsymbol{\Psi}^T \cdot \boldsymbol{Y}, \quad \boldsymbol{\Psi} = \frac{n_t}{n_y} \boldsymbol{S}_{UY} \cdot \left( \boldsymbol{U}_s \cdot \boldsymbol{U}_s^T \right)^{-1} \cdot \boldsymbol{U}_s \cdot \boldsymbol{Z}^T$$

To evaluate the skill of the model a crossvalidated hindcast is produced by applying the leave one year out methodology. In this way, the data corresponding to each year is omitted from the $\boldsymbol{Y}$ and $\boldsymbol{Z}$ matrices and MCA is applied with the remaining years. In each iteration, $\boldsymbol{\Psi}$ is calculated, as well as the predicted $\hat{\boldsymbol{Z}}$ for the omitted year, using the value of the predictor $\boldsymbol{Y}$ for

---

[1]In this case $A^i$ indicates the $i$-th row and $A_i$ the $i$-th column





that particular year. With the crossvalidated hindcast, the skill is evaluated by calculating the anomaly correlation coefficient (ACC) between $\boldsymbol{Z}$ and the crossvalidated $\hat{\boldsymbol{Z}}$. For each MCA iteration, the scf, $\boldsymbol{U}_s$ and $\boldsymbol{V}_s$, $\boldsymbol{S}_{UZ}$ and $\boldsymbol{S}_{UY}$ are stored in order to test the stability of the modes and its sensitivity for each particular year. In addition, the root mean squared error between the predictand, $\boldsymbol{Z}$, and the crossvalidated hindcast, $\hat{\boldsymbol{Z}}$, is calculated as:

$$\text{RMSE} = \sqrt{\frac{1}{n} \sum_{i=1}^{n} (z_i - \hat{z}_i)^2},$$

where n is the number of observations.

Validation is required for all models in order to test its reliability, assuming stationarity, i.e: the training period contains the same relations that the validation period. In this case the predictand field $\hat{\boldsymbol{Z}}$ is predicted calculating $\boldsymbol{\Psi}$ by applying the MCA for a certain training period and multiplying it by the predictor field $\boldsymbol{Y}$ for a validation period.

Although MCA has been applied to the same two case studies as in Suárez-Moreno and Rodríguez-Fonseca (2015) —The prediction of Pacific El Niño from the equatorial Atlantic and the prediction of Sahelian rainfall from the equatorial Pacific—the former will be used as an illustrative example throughout the paper. Additionally, the Sahelian prediction case is available in the zenodo folder (Duran-Fonseca and Rodriguez-Fonseca, 2024a).

The explanation of the model is done in section 3 through its application for the case study (listing 1-10).

The database chosen to define the predictor field is the sea surface temperature anomalies from HadISST database (Rayner et al., 2006). This data is available in monthly resolution. For the predictand field, HadISST is used as well. All the data and the code used in this paper together with the additional examples on Sahelian rainfall can be found in the zenodo folder (Duran-Fonseca and Rodriguez-Fonseca, 2024a).

## 3 Model Structure.

Spy4Cast is organized in three steps: setup, preprocess and methodology. The procedural workflow is illustrated in figure 1[2]. The complete configuration can be accomplished using the primary module `spy4cast`, while the preprocessing and methodology is done through the sub-module `spy4cast.spy4cast`.

The documentation for the package is published in Duran-Fonseca and Rodriguez-Fonseca (2023) and the code and data used in the paper is published in Duran-Fonseca and Rodriguez-Fonseca (2024a).

### 3.1 Configuration: loading data and slicing region.

Spy4Cast employs data-sets stored in the netCDF4 format (Wasser et al., 2021) and GRIB, a widely adopted standard in physics and data-science. Listing 1 shows how to create an instance of `Dataset` that allows for convenient dataset access and slicing operations on the data.

---

[2] `spy4cast.validation` is also a valid methodology but it works in a similar way as crossvalidation. Examples available in Duran-Fonseca and Rodriguez-Fonseca (2024a).



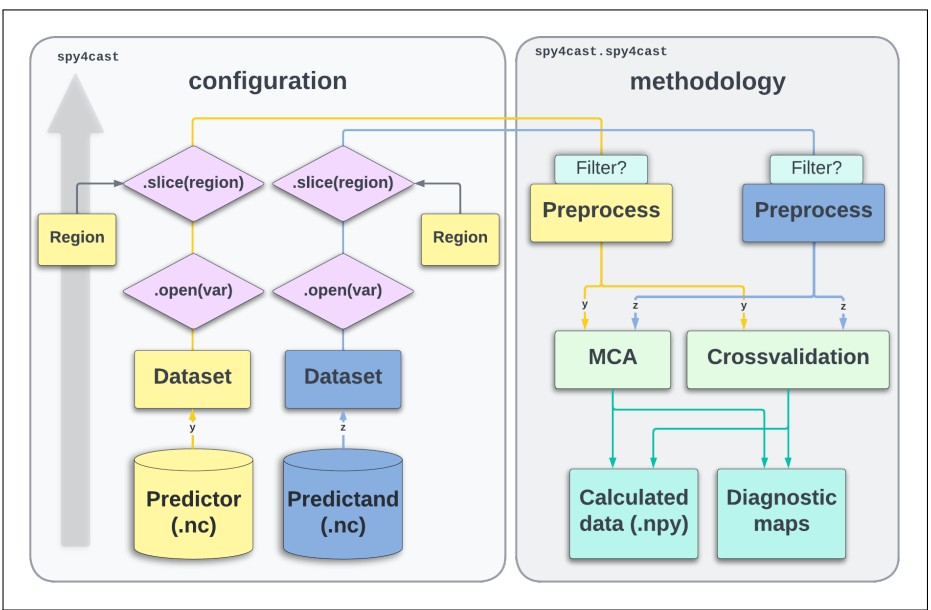

**Figure 1.** Flow Chart that describes the methodology configuration and application using Spy4Cast.

```python
import numpy as np  # Used later
from spy4cast import Dataset

dataset_folder = "./datasets"  # Path to the folder where the datasets are stored.
dataset_filename = "HadISST_sst-1970_2020.nc"  # File format must be netcdf4.
ds = Dataset(dataset_filename, folder=dataset_folder)
# A chunks keyword argument can be provided in this step. This value will be
# stored internally and passed to in the opening step to use dask chunks.
```

**Listing 1.** Python example to create an instance of `Dataset` to manipulate data in netCDF4 format.

The next step in configuration involves opening the data-set file with the `Dataset.open` method. This method will not load the data-set into memory, because it internally uses xarray function `xarray.open_dataset` (Hoyer and Hamman, 2017). Spy4Cast incorporates error-handling mechanisms to address potential issues after calling this function. One example is that in the event that `xarray.open_dataset` raises an exception due to an invalid time coordinate, `Dataset.open` has the capability to retry the data-set opening with `decode_time=False`. Subsequently, it intelligently generates a meaningful time variable based on the `units` attribute of the data-set (for now, this only works if the data is monthly). If all attempts prove unsuccessful, a `spy4cast.errors.DatasetError` exception is raised.

Spy4Cast centers around enhancing automation. Therefore, within `Dataset.open`, an effort is made to automatically identify the coordinate names for latitude and longitude. Acceptable coordinate names include `lat`, `latitude`, `lon`, or `longitude`. At the time of plotting maps, they can be shown in any region specified, not only centered in the Greenwich





meridian. `Dataset.open` requires a variable name if there are more than one variables in the data-set. Listing 2 shows how to open a dataset in this way for a variable called `sst`.

```
ds.open("sst")  # Opens the dataset, stores variables.
```

**Listing 2.** Python example to open a dataset with the variable `sst`.

To select the region and season to be analysed, `Region` and `Month` must be used together with `Dataset.slice`, that takes care of slicing the data-set into the area that the user wants to manipulate. The benefit of not loading the data-set into memory in the opening stage, is to avoid incorporating the entire data-set and only load the region that will be used for calculations. An intermediate interface is used for this step, called `Region`. Latitude must be specified from $-90°$ to $+90°$, where the positive make reference to degrees north. Longitude is specified from $-180°$ to $+180°$, where positive values

indicate degrees east to the Greenwich meridian. To help manage high-resolution data-sets, a `skip` argument can be passed into this function which indicates how many data points to skip in both dimension (latitude and longitude).[3]

```
from spy4cast import Region, Month

region = Region(
    lat0=-30, latf=30,
    lon0=-60, lonf=15,
    month0=Month.MAY, monthf=Month.JUL,
    year0=1976, yearf=2000
)  # months can also be stated through integers.
ds.slice(region)  # year0 and yearf apply to monthf.
# ds.slice(region, skip=1)  # skip 1 data point in lat and lon dimension.
```

**Listing 3.** Python example of slicing a data-set using the `Region` interface and `Dataset.slice`.

At this point it is interesting to have the chance of evaluating the correct selection of seasons and region. To this aim, `spy4cast.meteo` provides an easy way to visualize climate and anomaly maps, along with time series data for the chosen region. The images in figures 2 and 3 were created using the code provided in listing 4.

---

[3]The main purpose of `skip` is for the user to be able to run the methodology with lower resolution to make sure that the model is set up correctly. Once the user is satisfied with the results, the model can be run with the highest resolution just by setting the `skip` back to 0.



```python
from spy4cast.meteo import Clim, Anom

# Climatology maps and time series.
clim_map = Clim(ds, "map")  # Mean in the time dimension.
clim_ts = Clim(ds, "ts")  # Mean in the lat and lon dimension.
# Anomaly maps and time series
anom_map = Anom(ds, "map")  # An anomaly map for each year
anom_ts = Anom(ds, "ts")  # Mean in the lat and lon dimension
# Plot with the .plot method (look at docs). Example:
clim_map.plot(
    show_plot=True, save_fig=True,
    name="plots-Climatology_Anomaly_EquatorialAtlantic/clim_map.png",
    levels=np.arange(22, 28, 0.1),
    ticks=np.arange(22, 28.5, 0.5),
)
anom_map.plot(
    year=1997, show_plot=True, save_fig=True,
    name="plots-Climatology_Anomaly_EquatorialAtlantic/anom_map.png",
    levels=np.arange(-0.6, 0.6 + 0.05, 0.05),
    ticks=np.arange(-0.6, 0.8, 0.2),
)
# Save the data with the .save method (look at docs). Example:
anom_map.save("anom_map_", folder="./data-Climatology_Anomaly_EquatorialAtlantic/")
# Load previously saved data with the .load method (look at docs). Example:
anom_map = Anom.load("anom_map_", folder="./data-Climatology_Anomaly_EquatorialAtlantic/", type="map")
```

**Listing 4.** Python example to use the sub-package `spy4cast.meteo` to plot climatology and anomaly maps.

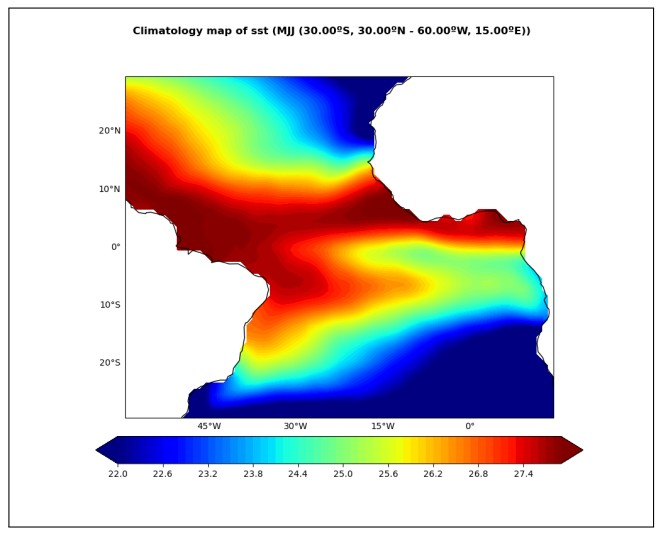

**Figure 2.** Climatology of sea surface temperature in the Equatorial Atlantic Ocean for the June to July season. Period: 1996 to 2019. The plot is generated with `spy4cast.meteo` (Listing 4)

## 165   3.2   Model Usage.

SpyCast acts upon a predictor and a predictand dataset. Once the predictand and predictor variables are selected, they need to be preprocessed before applying the MCA and crossvalidated hindcast. This preproccesing includes calculation of seasonal



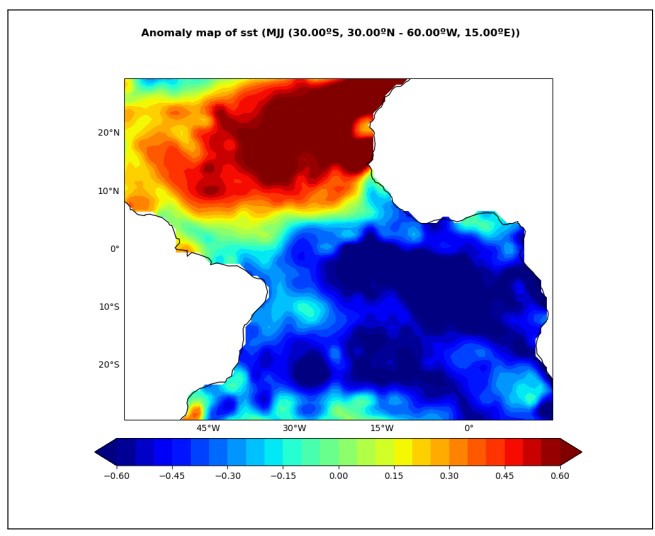

**Figure 3.** Sea surface temperature anomaly in the Equatorial Atlantic Ocean for the June to July season. Period 1996 to 2019. The plot is generated with spy4cast.meteo (Listing 4)

means, computation of seasonal anomalies and filtering. `spy4cast.spy4cast.preprocess` is a sub-module that is useful for other applications in climate variability studies and can be used independently. The outputs are organized in matrices with dimensions (space x time). Next, the MCA is applied, with a time linear-detrending of the data by default. Finally, a one-year leave out crossvalidation technique is applied. These three steps are part of the package `spy4cast.spy4cast.` All the data can be stored in `.npy` format (Harris et al., 2020). In the following subsection, a detailed explanation is provided for the correct execution of the model..

**Preprocess.**

As described earlier, Spy4CAST is encapsulated within the provided sub-module named `spy4cast.spy4cast.` To load and slice the predictor and predictand variables, follow the same steps as outlined in section 3.1 (listing 5).

The initial phase involves preprocessing, accomplished through the `Preprocess` interface. This interface is responsible for data preparation needed for Maximum Covariance Analysis (MCA), crossvalidation and validation. It performs the calculation of anomalies, and the application of time filtering algorithms. By employing the `Preprocess` class (as shown in listing 6), the computed three-dimensional matrices (time, latitude and longitude) are reshaped into space-time matrices. For long time series, various phenomena occurring at different frequencies may emerge. Therefore, the user can analyze the relationship between the predictor and the predictand by filtering out specific frequencies from the data. Thus, Spy4CAST can also apply a Butterworth filter adding th the `period` and `order` keyword arguments (Butterworth, 1930). The configuration and the preprocessing stage could be computationally intensive, and in certain instances, it might be more efficient to save the preprocessed matrices





```
# We will use sea surface temperature both for dataset and predictor, but
# with differnt regions
predictor = Dataset("HadISST_sst-1970_2020.nc", "./datasets").open("sst").slice(
    Region(lat0=-30, latf=10,
           lon0=-60, lonf=15,
           month0=Month.JUN, monthf=Month.AUG,
           year0=1970, yearf=2019),
    skip=1
)
predictand = Dataset("HadISST_sst-1970_2020.nc", "./datasets").open("sst").slice(
    Region(lat0=-30, latf=30,
           lon0=-200, lonf=-60,
           month0=Month.DEC, monthf=Month.FEB,
           # year0, yearf refer to monthf -> will span from DEC 1970 to FEB 2020
           year0=1971, yearf=2020),
    skip=1
)
#  There is a lag of 6 months (from June to December)
```

**Listing 5.** Python example of configuration a predictor and predictand data-set to input into Spy4Cast.

as `.npy` files with the method `Preprocess.save`. These files can be loaded directly using the `Preprocess.load` class
method, being able to skip the configuration step on next runs.

```
# First step. Preprocess variables: anomaly and reshaping
predictor_preprocessed = Preprocess(predictor, period=8, order=4)
predictor_preprocessed.save("y_", "./data-EquatorialAtalantic_Impact_Nino/")
# Save matrices as .npy for fast loading. To load use:
# predictor_preprocessed = Preprocess.load("y_",
#     "./data-EquatorialAtalantic_Impact_Nino/")
predictand_preprocessed = Preprocess(predictand)
predictand_preprocessed.save("z_", "./data-EquatorialAtalantic_Impact_Nino/")
# predictand_preprocessed = Preprocess.load("z_",
#     "./data/-EquatorialAtalantic_Impact_Nino")
```

**Listing 6.** Python example to preprocess the predictor and predictand data-sets configured in listing 5.

**MCA and Crossvalidation.**

Once the preprocessed data for $Y$ and $Z$ is computed, MCA is easily applied and only needs as arguments the predictor, the
predictand, the number of modes (`nm`) and the confidence level, `alpha` (listing 7) to be applied for assessing the significance
of the results. MCA uses `test-t` significance technique but the user can change it to `monte-carlo` non parametric test
easily, using random permutations from with `numpy.rand` (Harris et al., 2020). The results from the spatial configuration
of the modes are shown as homogeneous correlation maps for the predictor field and heteregeneous correlation maps for the
predictand field (see section Methodology and Data). These maps can be shown for the region in which MCA has been applied,
but also for an extended region in order to have a broader picture of the MCA mode using `map_y` and `map_z` as arguments to
`MCA.plot` (Duran-Fonseca and Rodriguez-Fonseca, 2024a, 2023). In addition, the expansion coefficients from the predictor





field (U) can be used to calculate other regression maps using different variables and datasets for the same period analysed (see `mca` and `index_regression` in Duran-Fonseca and Rodriguez-Fonseca (2024b)).

The user has two different ways of working with the computed outputs generated at this step. Firstly, the `MCA` object contains all attributes shown in figure 4 and, with packages like `matplotlib` (Hunter, 2007), custom figures can be produced.

Secondly, the computed matrices can be saved in `.npy` format using the `MCA.save`. These files can be easily loaded into any program, such as MATLAB, to produce figures according to the user's preferences.

The benefit of using `MCA.save` is that the user can rebuild the `MCA` instance with the output of the preprocessing stage (or using `Preprocess.load`) and with the class method `MCA.load`. In that way, they can be shared across without the need of rerunning the methodology of the previous steps.

```python
# Second step. MCA: expansion coefficients and correlation and regression maps
nm = 3
alpha = 0.05
mca = MCA(predictor_preprocessed, predictand_preprocessed, nm, alpha)
mca.save("mca_", "./data-EquatorialAtalantic_Impact_Nino/")
# mca = MCA.load("mca_", "./data-EquatorialAtalantic_Impact_Nino/",
#     dsy=predictor_preprocessed, dsz=predictand_preprocessed)
```

**Listing 7.** Python example to run the second step (MCA) in Spy4Cast for the predictor and predictand configured in listing 5.

Applied in a similar way to MCA, the third step, Crossvalidation, can be done directly in a few lines of code as illustrated in listing 8. To do this, only the preprocessed data from the predictor and predictand field and number of modes and significance level are required. The resulting matrices can be saved in `.npy` format and loaded as explained previously for being further analysed by the user. The name of the accessors for each of the resultant matrices can be consulted in figure 4.

```python
# Third step. Crossvalidation: skill and hidcast evaluation and products
cross = Crossvalidation(predictor_preprocessed, predictand_preprocessed, nm, alpha)
cross.save("cross_", "./data-EquatorialAtalantic_Impact_Nino/")
# cross = Crossvalidation.load("cross_", "./data-EquatorialAtalantic_Impact_Nino/",
#     dsy=predictor_preprocessed, dsz=predictand_preprocessed)
```

**Listing 8.** Python example to run the third step (crossvalidation) in Spy4Cast for the predictor and predictand configured in listing 5.

A way to visualize the results with a minimal extra code is using the `.plot` method of `Preprocess`, `MCA` and
`Crossvalidation`. Figures 5 and 6 were created with the code from listing 9 that takes advantage of this fast plotting method. Arguments `map_y` and `map_z` were used to create a global regression along a larger region. For each plotting functions there are special arguments and settings that can be consulted in the documentation (Duran-Fonseca and Rodriguez-Fonseca, 2023).





```
map_y = Preprocess(Dataset("HadISST_sst-1970_2020.nc", "./datasets").open("sst").slice(
    Region(lat0=-80, latf=50, lon0=-180, lonf=50,
        month0=Month.JUN, monthf=Month.AUG,
        year0=1970, yearf=2019), skip=1))
map_z = Preprocess(Dataset("HadISST_sst-1970_2020.nc", "./datasets").open("sst").slice(
    Region(lat0=-70, latf=70, lon0=-300, lonf=-40, month0=Month.DEC, monthf=Month.FEB,
    year0=1971, yearf=2020), skip=1))
mca.plot(save_fig=True, name="mca_global.png",
        map_y=map_y,
        map_z=map_z,
        figsize=(13, 8),
        width_ratios=[1.5, 2, 2.1],
        folder="./plots-EquatorialAtalantic_Impact_Nino/",
        ruy_ticks=[-1, -0.5, 0, 0.5, 1], ruz_ticks=[-1, -0.5, 0, 0.5, 1])
cross.plot(save_fig=True, name="cross.png",
        folder="./plots-EquatorialAtalantic_Impact_Nino/")
cross.plot_zhat(1998, figsize=(12, 10), save_fig=True, name="zhat_1998.png",
            folder="./plots-EquatorialAtalantic_Impact_Nino/",
            z_levels=np.linspace(-2, 2, 10), z_ticks=np.linspace(-2, 2, 5))
```

**Listing 9.** Python code to show debug plots for MCA and Crossvalidation results produced in section 3.2.

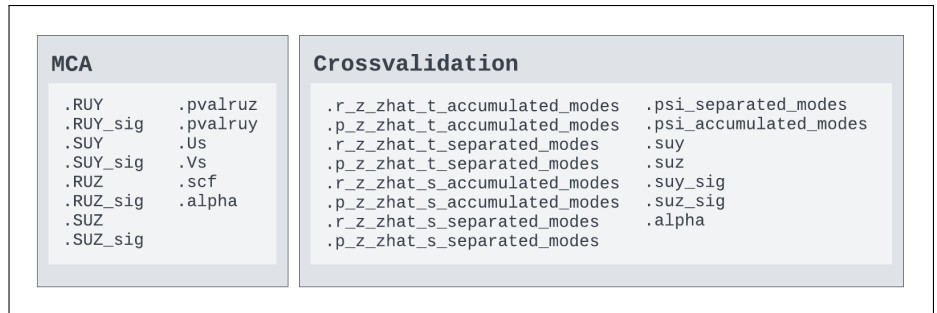

**Figure 4.** Name of the accessors for each result array of MCA and Crossvalidation.

**Validation.**

Spy4Cast is able to perform a validation methodology to look for non-stationary relations. To run validation, four datasets need to be preprocessed, corresponding to the predictor and predictand field, for the training period and for the validating period. With the training dataset, MCA is applied to produce the prediction model parameters. In this way, the $\Psi$ matrix is calculated for the MCA modes characteristic of the training period (see section Methodology and Data). Then, using $\Psi$ and the preprocessed predictor field for the validation period, the validated predictand field is calculated . Thus, the model is

tested against a predictor and predictand field of a different period than the training one. Listing 10 shows an example of the application of `spy4cast.Validation` to the example before, by training on data from 1976 - 2000 and validating on the period from 2001 - 2019. Figure 7 shows the output of the correlation in space and time between the predictor and predicting field. The results show that there is a non stationarity in the relation during that period due to the poor correlation between the training and those validated fields. This is an important issue to be taken into account when applying this methodology, due to the fact that the non stationarity of the teleconnections is a hot spot in climate variability studies (Rodríguez-Fonseca et al.,



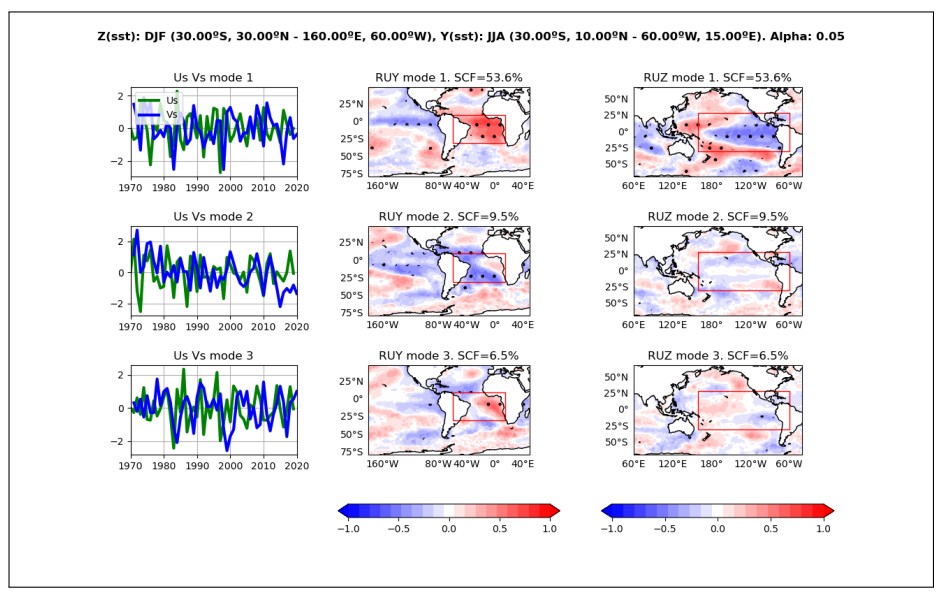

**Figure 5.** Fast plot of `MCA` output predicting Niño with the sea surface temperature of the equatorial Atlantic Ocean with a lag of 7 months from 1977 to 2001.

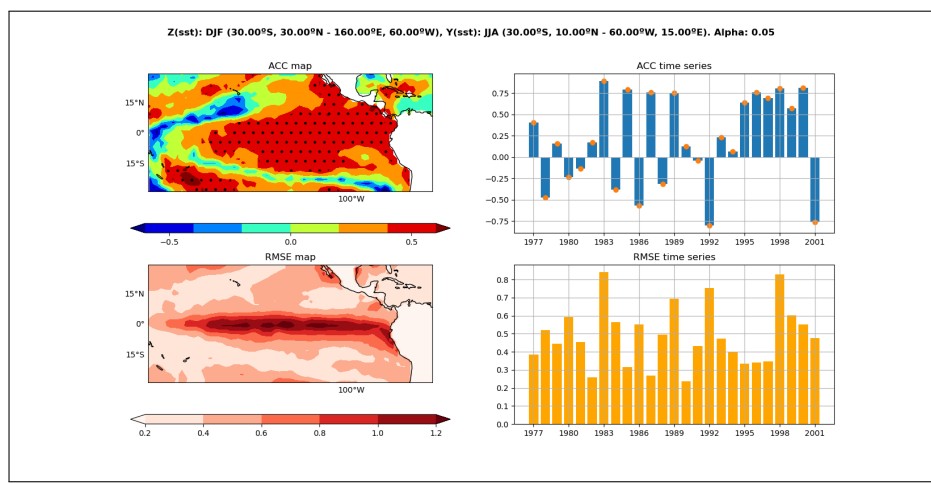

**Figure 6.** Fast plot of `Crossvalidation` output predicting Niño with the sea surface temperature of the equatorial Atlantic Ocean with a lag of 7 months from 1977 to 2001.

2016) . Using spy4cast, sensitivity experiments could be defined, training with different periods to identify those validating periods which follow the same modes of covariability.

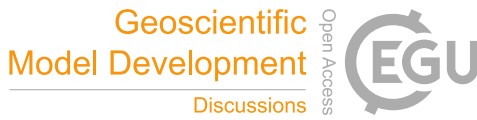

```python
# To apply validation we first preprocess the training data
training_y = Preprocess(Dataset("HadISST_sst-1970_2020.nc", "./datasets").open("sst").slice(
    Region(lat0=-30, latf=10, lon0=-60, lonf=15, month0=Month.JUN, monthf=Month.AUG,
        year0=1970, yearf=2000), skip=1))
training_z = Preprocess(Dataset("HadISST_sst-1970_2020.nc", "./datasets").open("sst").slice(
    Region(lat0=-30, latf=30, lon0=-200, lonf=-60, month0=Month.DEC, monthf=Month.FEB,
        year0=1971, yearf=2001), skip=1))
training_mca = MCA(training_y, training_z, nm=6, alpha=0.05)
# We now validate agains the period from 2001 - 2020
validating_y = Preprocess(Dataset("HadISST_sst-1970_2020.nc", "./datasets").open("sst").slice(
    Region(lat0=-30, latf=10, lon0=-60, lonf=15, month0=Month.JUN, monthf=Month.AUG,
        year0=2010, yearf=2019), skip=1))
validating_z = Preprocess(Dataset("HadISST_sst-1970_2020.nc", "./datasets").open("sst").slice(
    Region(lat0=-30, latf=30, lon0=-200, lonf=-60, month0=Month.DEC, monthf=Month.FEB,
        year0=2011, yearf=2020), skip=1))
validation = Validation(training_mca, validating_y, validating_z)
validation.plot(save_fig=True, folder="./plots-EquatorialAtalantic_Impact_Nino/", name="validation.png",
                version="default")
```

**Listing 10.** Python code to run validation.

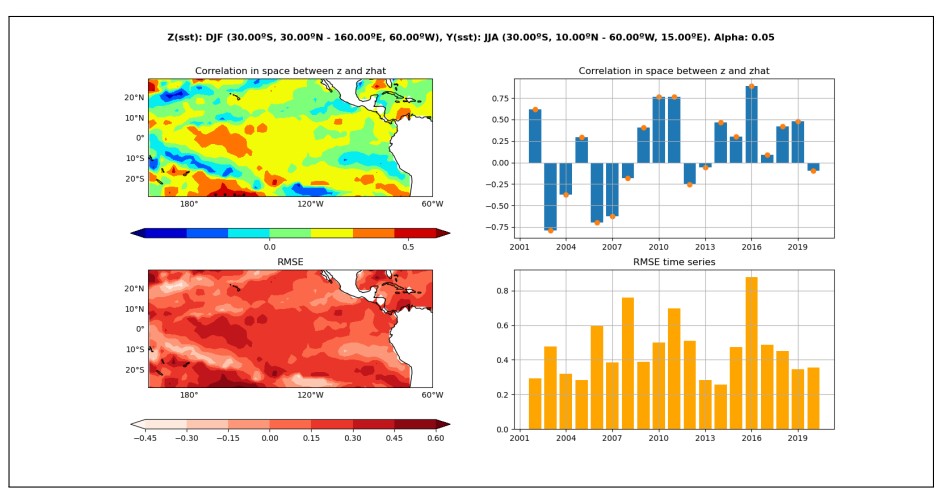

**Figure 7.** Fast plot of `Validation` output predicting SST anomalies in El Niño region using sea surface temperature of the equatorial Atlantic Ocean with a lag of 7 months as predictor field. The training period is 1977-2001 and the validating period is 2001-2019.

## 4 Application to the study the impact of the Atlantic Impact on the Pacific.

Along the present paper, the spy4cast methodology has been applied for illustrating a state of the art teleconnection pattern
which was discovered in early 2000's: the tropical Atlantic impact on ENSO (Polo et al., 2008; Rodríguez-Fonseca et al., 2009;
Ding et al., 2012; Cai et al., 2019). This teleconnection pattern identifies the so-called 'Atlantic Niño' as a predictor of ENSO
. The Atlantic Niño is a mode of variability characterized by sea surface temperature anomalies in the equatorial Atlantic,
with dynamics similar to those of the Pacific Niño (Lübbecke et al., 2018). During the warm phase of this mode, increased
convection and an alteration of the Walker cell occur. This alteration in the Walker cell leads to a change in the trade winds over
the western equatorial Pacific. This change in winds triggers oceanic Kelvin waves that initiate a La Niña episode (Polo et al.,



2015). Several authors have shown how this Atlantic phenomenon, which peaks in the boreal summer, can be used to predict the occurrence of an ENSO event six months later (Keenlyside et al., 2013; Martín-Rey et al., 2015; Exarchou et al., 2021). ENSO is the interannual variability mode with the greatest influence on global climate at interannual scales. Predicting this phenomenon has significant implications due to its importance. However, this teleconnection between the Atlantic and Pacific

is non-stationary and varies at multidecadal time scales (Martín-Rey et al., 2014; Kucharski et al., 2016).

All these features described above are well represented using spy4cast. In this way, the period 1977-2001 is chosen to create the predictand and predictor fields from SST anomalies in the tropical Atlantic (in june-july-august, JJA) and Pacific basins in the next december-january-february (DJF) , preprocessing them (listing 5, 6). After running the MCA and crossvalidation, different variables are created (see figure 4). Spy4cast identifies the Atlantic Niño in JJA as the main mode of covariability with

DJF Pacific SST anomalies (listing 7). The heterogeneous correlation map of the leading mode resembles La Niña (El Niño) pattern for positive (negative) values of the expansion coefficient Us. This expansion coefficient is highly correlated with all grid points in the equatorial Atlantic, shaping the Atlantic Niño fenomenon. This leading mode explains almost the 60 % of the covariability . The rest of modes are not related to the equatorial part of ENSO, but to other regions in the Pacific, which can provide predictability to the Pacific. Using the first 3 modes, almost the 76 % of the covariance is explained. These modes are

used to produce a crossvalidated hindcast, which spatial and temporal skills are represented in Figure 6 (top panel). On the one hand, all the equatorial region of the Pacific and some off-equatorial regions in the western flank present high scores of the ACC indicating the potential of the equatorial Atlantic to predict the whole spatial structure of the ENSO. On the other hand, almost all the years present high skill values. Those with no skill correspond to years in which no Atlantic Niño (Niña) occurs (see the expansion coefficient in Figure 5). In the lower panel of Figure 6 the RMSE is plotted, showing the highest values in the

equatorial region, which is the region in which the largest anomalies also occur. Some high RMSE values correspond to years where the prediction is not good. For example, in 1992 the prediction is not good (low ACC) but also the RMSE is high. This fact does not hold for the whole period, suggesting that the linear nature of this methodology cannot always produce accurate predictions, as there are other non-linear relationships that the MCA is not able to capture. Regardless of the limitations of this methodology, spy4cast is a convenient tool to assess the existance of teleconections and the reliability of the predictions, as

well as their stationarity.





## 5 Conclusion and Discussion

Spy4Cast is a powerful tool for manipulating climate datasets and implementing Maximum Covariance Analysis (MCA) for both seasonal and decadal prediction applications. This API enhances the automation and has proven effective in increasing productivity and the quality of research.

Spy4Cast represents the beginning of a new approach to statistical seasonal forecasting and the analysis of key aspects, such as the non-stationarity of teleconnections. Although the core of the model is based on classical Maximum Covariance Analysis (MCA), this API is more versatile, featuring a data preprocessing module that can be used for various other applications. Additionally, it offers a module for visualizing and storing results, which can be utilized productively.

Moreover, this API can be coupled with large databases, such as those from the Coupled Model Intercomparison Project 270    (CMIP). Indeed, within the OFF project, it is being integrated into ESMValTool (Righi et al., 2020) to evaluate changes in teleconnections and predictability under future scenarios. ESMValTool (Earth System Model Evaluation Tool) is a software package designed to facilitate the evaluation and analysis of Earth System Models (ESMs), allowing the incorporation of Python scripts into the analysis.

Spy4Cast is still in development. Future versions will also include the capability to replace the prediction algorithm with 275    other machine learning techniques. Spy4Cast is open to improvement, and all suggestions are welcome. These can be submitted to the authors or through the GitHub platform (Duran-Fonseca, 2023).

*Code and data availability.*    The source code for Spy4Cast is located in the zenodo folder at https://doi.org/10.5281/zenodo.14017545 under Apache License. The code and data used as examples in the paper in the zenodo folder at https://doi.org/10.5281/zenodo.13619197.

*Author contributions.*    The programming of the model's code, the documentation and the distribution has been done by Pablo Duran-Fonseca. 280    The design of the methodology, the fundamental functions, and the practical applications have been carried out by Belén Rodríguez-Fonseca.

ther geographical representation in this paper. While Copernicus Publications makes every effort to include appropriate place names, the final responsibility lies with the authors.

*Competing interests.*    No competing interests are present.





*Acknowledgements.* This research has been supported by the European Union NEXTGEMS (grant agreement 101003470) and TRIATLAS (grant agreement 817578) projects and the Spanish DISTROPIA (PID2021-792 125806NB-I00) and OFF (TED2021-130106B-I00) projects.



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
