# Peer review of "Spy4Cast v1.0: a Python Tool for statistical seasonal forecast based on Maximum Covariance Analysis"

_Geoscientific Model Development, 2024_

## Referee Comment (RC1)

**Review of**
**"Spy4Cast v1.0: a Python Tool for statistical seasonal forecast based on Maximum Covariance Analysis"**

Pablo Duran-Fonseca and Belén Rodríguez-Fonseca

January 29, 2025

**General Comments**

This manuscript presents *Spy4Cast*, a Python-based library aimed at facilitating Maximum Covariance Analysis (MCA) in climate research, especially for seasonal prediction and teleconnection studies. While the topic is highly relevant to *Geoscientific Model Development* and the example of the Atlantic Niño—ENSO teleconnection demonstrates the tool's potential, several core issues must be addressed to meet GMD standards:

1. **Scientific Context and Justification:** While the manuscript provides a thorough description of MCA, it does not sufficiently position MCA within the broader landscape of statistical climate forecasting methods, such as Canonical Correlation Analysis (CCA). Given that CCA maximizes correlation—often the primary metric in forecast validation—it is important to clarify why MCA was chosen over CCA and in which scenarios it is preferable or less suitable. Additionally, *Spy4Cast* should be contextualized within the existing Python ecosystem for climate forecasting and analysis (e.g., XCast, xeofs, climpred), which similarly leverage xarray/dask. A discussion of the conceptual and practical advantages of *Spy4Cast* compared to these tools—such as what *Spy4Cast* offers that XCast does not—would strengthen the manuscript (see Specific Comments 3 and 8).

2. **Scalability and Computational Efficiency:** The manuscript states that *Spy4Cast* can handle "large" climate datasets, but its reliance on in-memory NumPy arrays for computing the cross-covariance matrix raises concerns about scalability. High-resolution or global-scale datasets could result in excessive memory usage, potentially limiting the tool's applicability to gigabyte- or terabyte-scale datasets. The authors should clarify the realistic data-size limits of *Spy4Cast* and discuss potential strategies for improving scalability, such as incorporating `dask.array`, processing data in smaller subregions, or applying prior dimensionality reduction techniques like PCA (see Specific Comment 1).

3. **Scope and Flexibility:** The manuscript suggests that *Spy4Cast* can handle "any kind of predictor," yet the current implementation appears tailored to oceanic fields with latitude-longitude-time coordinates. While the library's specialized design provides advantages, it also seems to come at the cost of flexibility—for example, through the use of custom classes such as `Dataset` and `Region`. It would be helpful to clarify whether the tool can process land-based variables, vertical levels (e.g., depth or pressure), or multiple variables simultaneously (see Specific Comments 21, 22, and 25).

4. **Clarity, Structure and Documentation:** Although *Spy4Cast* is openly licensed and includes example scripts, the documentation remains sparse, requiring users to infer key details from the source code. The manuscript refers to preprocessing steps (e.g., detrending, filtering) but does not fully explain their implementation. Additionally, some design choices—such as the use of specialized `Dataset` and `Region` classes instead of a pure xarray-based approach—should be more clearly justified, with discussion of their advantages and limitations. The structure of the manuscript would also benefit from a clearer distinction between the two validation approaches used: historical split validation and leave-one-year-out cross-validation. Finally, a more comprehensive user manual—including details on parameter options, algorithmic considerations (e.g., the use of randomized SVD), and potential pitfalls—would enhance reproducibility and usability (see Specific Comments 2,18,19,20,29,24,28, and 27).

In summary, *Spy4Cast* shows promise for moderate-scale, MCA-based forecasting research. However, its impact would be strengthened by (i) a broader discussion of MCA in relation to other statistical methods, (ii) a clear assessment of its current scope and scalability limitations, and (iii) improved presentation and documentation, either within this manuscript or through comprehensive online resources.

**Specific Comments**

**Comment 1 (L7; also L48-53, 67, 69)**

The manuscript states that *Spy4Cast* "enables large dataset manipulation," yet the internal computations appear to rely on in-memory NumPy arrays for constructing the cross-covariance matrix. To clarify the tool's practical limits, please specify:

- The maximum feasible input size (e.g. grid dimensions) before memory constraints become prohibitive.

- Whether *Spy4Cast* integrates with `dask.array` or chunked xarray computations, specifically the computation of the SVD. If not, consider explicitly stating that *Spy4Cast* is suited for moderate-scale datasets but may not efficiently handle very large (gigabyte-scale and beyond) datasets.

Additionally, please revisit the use of the term "scalability" (L69). While static type-checking improves code maintainability, it does not directly enhance the tool's ability to handle large datasets. Consider rephrasing for accuracy.

**Comment 2 (L13)** *...well documented for beginners and experience programmers.*

The statement about the documentation being suitable for both beginners and experienced programmers may need reconsideration. The current documentation appears quite minimal—while one notebook demonstrates the general workflow, there is little to no explanation of the individual preprocessing steps and their implications. Expanding the documentation to provide clearer descriptions of these steps would greatly enhance usability.

**Comment 3 (L16-40)**

The manuscript provides a detailed discussion of MCA, while Canonical Correlation Analysis (CCA) is mentioned only briefly. Since many operational forecasting systems rely on CCA—given that it maximizes correlation, which often aligns more directly with forecast evaluation criteria—please expand on the rationale for focusing on MCA.

Specifically, clarify the trade-offs between MCA and CCA in statistical forecasting. Additionally, consider citing or discussing more recent literature (e.g., [1]) that compares these methods.

If feasible, a brief note on whether Spy4Cast could be extended or adapted for CCA—or if it is specifically designed for MCA—would be valuable.

**Comment 4 (L21 onward)**

The manuscript consistently refers to one field as the "predictor" and the other as the "predictand," though MCA itself is inherently symmetric and does not imply a directional causal relationship. To avoid potential misunderstanding, consider clarifying that this terminology is used specifically in the context of forecasting rather than as a fundamental property of MCA.

**Comment 5 (L21)** *In this context, MCA analysis provides spatial patterns of the predictor and predictand field which are related by teleconnections.*

The text suggests that all MCA modes inherently represent teleconnections or causal relationships. Please clarify that these modes primarily capture statistical covariance and do not necessarily indicate causal links.

**Comment 6 (L33)** *...it has the advantage of being easily interpretable ...*

Higher-order MCA modes can become increasingly difficult to interpret due to the orthogonality constraints of SVD, potentially leading to so-called "Buell patterns" (similar to PCA) [e.g., 2]. Please acknowledge this limitation when discussing the interpretation of the second or third MCA modes.

**Comment 7 (L41)** *...a new paradigm of research in climate variability studies has emerged ...*

This statement is vague for readers unfamiliar with Cai et al. (2019). Please specify what this paradigm entails (e.g., cross-basin interactions, trans-basin teleconnections) and clarify how *Spy4Cast* relates to it.

**Comment 8 (L48-53)**

The manuscript would benefit from a more thorough discussion of other Python-based forecasting and dimensionality-reduction packages, such as XCast [3], climpred [4], and xeofs [5]. Please compare *Spy4Cast*'s approach, strengths, and limitations relative to these tools—for example, in-memory vs. distributed computing or a specialized feature set vs. a more general framework. Expanding this discussion would help clarify *Spy4Cast*'s positioning within the broader climate data analysis software ecosystem.

**Comment 9 (L83)**

The definition of the cross-covariance matrix assumes zero-mean time series. Additionally, the formulation appears to be missing a normalization factor of $1/n_t$ or $1/(n_t - 1)$ for an unbiased covariance estimation. Please clarify or correct this as needed.

**Comment 10 (L89)** *The information of this matrix is redundant . . .*

It would be more precise to describe this as a high degree of redundancy—or even better, as multicollinearity.

**Comment 11 (L89)** *. . . produce the same maps . . .*

This statement is generally incorrect. You likely mean "similar maps" rather than identical ones. The claim would only be true if two time series were exactly equal, which is unlikely in practice.

**Comment 12 (L89)** *Also it is a complex matrix as it takes into account all possible relations between points.*

Are you certain this includes *all* possible relationships, or only linear ones? Consider revising for accuracy. Additionally, rather than making this statement, it may be more useful to emphasize that the matrix grows rapidly in size, with dimensions $n_y \times n_z$.

**Comment 13 (L94)** *. . . matrix of eigenvalues . . .*

The result of SVD provides a diagonal matrix containing the singular values, not eigenvalues. Please correct this terminology.

**Comment 14 (L94)** *. . . which represents the squared covariance fraction . . .*

Each singular value represents the covariance explained, while the squared singular values correspond to the squared covariance. The Squared Covariance Fraction (SCF) can be computed from the singular values, but it is not provided directly.

**Comment 15 (L100)** *. . . fraction of variance . . .*

fraction of **squared co**variance

**Comment 16 (L101)** *. . . which are linked by having maximum covariance.*

More precisely, the first mode is linked by maximum covariance. Subsequent modes follow these principles: (i) Mode $i + 1$ captures the maximum covariance of the remaining data after the first $i$ modes have been removed. (ii) This is subject to the constraint that the patterns $R_{i+1}$ and $Q^{i+1}$ are orthogonal.

**Comment 17 (L120)** *. . . where n is the number of observations.*

I assume that $n$ refers to the number of grid points or spatial locations. The term "observations" may imply a temporal scale, which does not seem to be the case here. Please clarify.

**Comment 18 (L134-135)** *Spy4Cast is organized in three steps: setup, preprocess and methodology. The procedural workflow is illustrated in figure 1.*

The text appears to contradict Figure 1, which explicitly separates configuration and methodology, while treating preprocessing as part of methodology (see Specific Comment 46). Please clarify or revise for consistency.

**Comment 19 (L135-136)**

Figure 1 presents a logical workflow, yet this structure does not appear to be reflected in the design of the software. Could you clarify the reasoning behind this discrepancy?

**Comment 20 (L139)** *Configuration: loading data and slicing region.*

I would argue that loading data and slicing a region are distinct from configuration. Shouldn't configuration be limited to defining metadata, with data loading and preprocessing occurring separately based on that configuration? Merging these logically different concepts seems confusing. Could you clarify the reasoning behind this design choice?

**Comment 21 (L140-165)**

It seems that some functionalities, such as temporal and spatial slicing, are already available in xarray. Could you clarify why you chose to introduce custom classes like `Dataset` and `Region` instead of relying on xarray's native methods?

Additionally, does *Spy4Cast* support advanced xarray features like `open_mfdataset`, which are essential for working with climate datasets distributed across multiple files?

Finally, for users who need custom preprocessing steps (e.g., weighting, detrending), can they seamlessly revert to or integrate with standard xarray workflows?

**Comment 22 (L140-165)**

Currently, *Spy4Cast* appears to support only 2D or 3D ocean-centric data (longitude, latitude, time). If the intention is to allow "any kind of predictor" (L9,10), please clarify whether different variables (e.g., global soil moisture) or multi-dimensional data (e.g., depth, forecast members) can be seamlessly incorporated.

Additionally, do the constraints for the predictor apply equally to the predictand?

If the current implementation is specialized (e.g., limited to ocean-only fields or requiring specific naming conventions), please make these limitations explicit in the user documentation.

**Comment 23 (L144)** *This method will not load the data-set into memory, because it internally uses xarray function* `xarray.open_dataset`

`xarray.open_dataset` does not inherently prevent loading the dataset into memory. The key factor is whether the `chunks` argument is specified. Please clarify this point.

**Comment 24 (L168-179)** *This preproccesing includes calculation of seasonal means, computation of seasonal anomalies and filtering. [...] Next, the MCA is applied, with a time linear-detrending of the data by default.*

The methods for time filtering and linear detrending require more detail:

- Is the linear detrending applied gridpoint-wise, or across the monthly or seasonal time series?
- What does "frequency filtering" entail (e.g., Butterworth filter parameters, cutoff frequencies)?
- How is missing data handled (e.g., masked land areas)?

Consider including these details in the "Preprocessing" section for clarity.

Additionally, if *Spy4Cast* is primarily designed for ocean variables, please specify how continental points and other 3D dimensions (e.g., depth or altitude) are treated.

**Comment 25 (L172)** *All the data can be stored in* `.npy` *format.*

Storing preprocessed data in `.npy` format may reduce compatibility with xarray/dask workflows and is less common for data exchange within the climate science community. Could you clarify the rationale for choosing `.npy` over standard formats like netCDF or Zarr, which are widely used for large-scale climate data? If this choice is primarily for internal convenience, please state that explicitly.

**Comment 26 (L187)** *MCA and Crossvalidation*

It would be clearer to separate MCA and cross-validation into two distinct sections. In the MCA section, please specify that you use approximate algorithms based on randomized linear algebra to accelerate singular value decomposition and provide relevant references.

**Comment 27 (L190)** *MCA uses test-t significance technique ...*

How is the t-test applied—one-sided or two-sided? What is the target variable? I assume it is the correlation coefficients of the homogeneous/heterogeneous correlation patterns. Additionally, please account for multiple testing when interpreting p-values or discuss how this issue is addressed.

**Comment 28 (196)** *. . . can be used to calculate other regression maps using different variables and datasets for the same period analysed (see* `mca` *and* `index_regression` *in Duran-Fonseca and Rodriguez-Fonseca (2024b)).*

Please clarify this statement in more detail. Referring readers to the `mca` and `index_regression` submodules of *Spy4Cast* without further explanation makes it difficult to understand the intended meaning without digging through the code. Providing a brief description of how these regression maps are computed would improve clarity.

**Comment 29 (L211)** *Arguments* `map_y` *and* `map_z` *were used to create a global regression along a larger region.*

Please elaborate on the meaning of `map_y` and `map_z`. What exactly do these arguments represent? Additionally, the phrase "global regression along a larger region" is unclear—a larger region is not necessarily global. Could you clarify what is being regressed against what?

**Comment 30 (L205-225)**

The manuscript would benefit from a more structured explanation of validation and cross-validation:

- I suggest consolidating both discussions into a single section, as they are closely related. While cross-validation employs classical leave-one-out cross-validation (LOO-CV), the validation approach instead partitions the dataset into concurrent (historical) time slices to account for long-term temporal autocorrelation.

- Additionally, please clarify the apparent contradiction between "Spy4Cast is not designed to assess stationarity" (L56) and "Spy4Cast is able to [...] look for non-stationary relations" (L215). Clearly defining the tool's actual capabilities and limitations in this context would improve consistency.

**Comment 31 (L226,227)**

I find this sentence difficult to understand (apologies, as I am not a native English speaker). Could you rephrase it to clarify the intended meaning?

**Comment 32 (L229-240)**

To provide broader context and strengthen the demonstration of *Spy4Cast*'s applicability, consider incorporating recent references on Atlantic Niño weakening under climate change (e.g., Crespo *et al.* [6]). This would further highlight the tool's relevance to current climate research questions.

**Comment 33 (L241)** *. . . all these features are well represented by spy4cast.*

Could you clarify what is meant by this statement? For example, earlier, you mention that the Atlantic phenomenon peaks in boreal summer, but this appears to be an a priori choice in the modeling process rather than an outcome inherently produced by the software.

Do you mean that these features can be accommodated by the researcher when using *Spy4Cast*, implying that the key strength is its flexibility? If so, please rephrase for clarity.

**Comment 34 (L244)** *. . . different variables are created.*

Please specify which variables are created. Are these evaluation metrics, MCA results, or something else? Figure 4 only displays file names, making it difficult to determine this with certainty.

**Comment 35 (L244)** *Spy4cast identifies the Atlantic Niño in JJA as the main mode of covariability with DJF Pacific SST anomalies.*

While *Spy4Cast* does identify the Atlantic Niño in JJA as the main mode of covariability with DJF Pacific SST anomalies, this result is largely determined by the specific choice of seasons and regions (tropical Pacific and tropical Atlantic) in the analysis. Given these constraints, what alternative outcomes could have emerged? Would the result change if the predictor region were expanded?

In the introduction, you mention that MCA can help guide more advanced prediction algorithms by identifying potential predictor regions. However, your case study presupposes prior knowledge of the predictor region. Would it be more

aligned with your introduction to assume little to no a priori knowledge and let MCA reveal potential predictors (e.g., Atlantic El Niño)?

This is not necessarily a suggestion for the revised manuscript, but rather a potential idea for future exploration.

**Comment 36 (L246-247)** *This expansion coefficient is highly correlated with all grid points in the equatorial Atlantic, shaping the Atlantic Niño phenomenon.*

Please provide a quantitative measure to support this statement. For example, you could compute the Pearson correlation coefficient between the expansion coefficients and the Oceanic Niño Index (ONI) to quantify their similarity to ENSO.

**Comment 37 (L247)** *This leading mode explains almost the 60 % of the covariability*

60 % of the **squared** covariance.

**Comment 38 (L249)**

**squared** covariance

**Comment 39 (L257-258)** *This fact does not hold for the whole period, suggesting that the linear nature of this methodology cannot always produce accurate predictions, as there are other non-linear relationshps that the MCA is not able to capture.*

You could note that a discrepancy between different evaluation metrics (e.g., high ACC but high RMSE) often indicates a bias in the error distribution. This bias may stem from the inherent linear assumptions of MCA, which, as you correctly point out, can limit its ability to predict nonlinear extreme events.

**Comment 40 (L263)** *This API [. . . ] has proven effective in increasing productivity and the quality of research.*

Unless there is concrete evidence supporting this claim, I would suggest softening the statement. Instead, you could say that the API has the potential to improve productivity and reproducibility in research.

**Comment 41 (L265)** *Spy4Cast represent the beginning of a new approach to statistical seasonal forecasting . . .*

What exactly do you mean by "new approach"? MCA itself is not new—are you referring to the use of predefined routines for analysis? If so, this is also not entirely novel, as operational forecast centers routinely employ such methods for statistical seasonal forecasting. To ensure accuracy, consider avoiding broad claims about "new approaches" unless specific evidence or metrics are provided.

**Comment 42 (L267)** *. . . this API is more versatile . . .*

Please clarify what makes this API "more versatile" compared to existing open-source solutions. Providing specific examples or comparisons would help substantiate this claim.

**Comment 43 (L270)** *Indeed, within the OFF project, it is being integrated into ESMValTool.*

Could you provide more details on the planned integration with ESMValTool, such as the expected timeline and intended functionality? Additionally, if "OFF" refers to a specific project, please define the acronym for clarity.

**Comment 44 (Conclusion and Discussion)**

Consider separating the discussion of the tool's limitations and future directions from the concluding remarks to follow a more standard "Conclusion and Outlook" structure.

**Comment 45 (Listings 1-10)**

The numerous short code listings may quickly become outdated if the API changes. Consider moving them to an online supplement or user guide while keeping only a concise set of essential examples in the main text. This would help maintain the paper's focus while ensuring comprehensive examples remain accessible in the documentation. Additionally, reassess whether all listings are necessary in their current form—Listing 2, for instance, provides limited information.

**Comment 46 (Figure 1)**

Could you clarify the meaning of the left-hand-side arrow? Additionally, there appears to be an inconsistency between the manuscript (L134) and Figure 1 regarding the workflow structure. The text describes three steps: configuration,

preprocessing, and methodology (which includes MCA and validation). However, in the figure, the workflow is grouped into only two categories: configuration and methodology, with preprocessing included under methodology. I would argue that temporal and spatial slicing operations are also part of preprocessing.

Please explain and justify why the workflow is structured this way. Additionally, the different abstraction levels implied by the colors, boxes, and shapes are somewhat unclear. A more explicit explanation of how these visual elements correspond to the workflow's logical structure would improve clarity.

**Comment 47 (Figure 4)**

This figure is difficult to interpret without additional context. If it is meant as a quick reference, consider adding a brief explanation in the caption about the typical use and content of each array. Otherwise, reassess whether the figure is essential for the manuscript's long-term clarity and sustainability (cf. Specific Comment 45).

**Comment 48 (Figure 5)**

It appears that this figure presents the output of the MCA (expansion coefficients and homogeneous/heterogeneous correlation patterns for modes 1 to 3), yet the caption states "predicting Niño." However, no actual prediction is shown—only the covarying patterns of variability between time-lagged SST in the tropical Pacific and Atlantic.

**Comment 49 (Figure 6)**

When presenting ACC and RMSE, please clarify the temporal dimension over which they are computed and specify the reference variable or index. Additionally, provide units for RMSE (presumably °C).

It would also be helpful to briefly discuss the error in the context of SST variability—is the RMSE relatively low or large compared to typical SST variations?

**Comment 50 (Figure 6)**

Please provide a more detailed description in the caption. Specifically, what do the orange dots in the upper-right panel represent? Do they indicate uncertainties in ACC values? If so, how are these uncertainties calculated?

**Comment 51 (Figure 7)**

RMSE is typically non-negative. Could you clarify how a negative RMSE appears in the figure?

**Comment 52 (Figures–General)**

- Some figures (particularly Figures 2, 3, 6, and 7) appear to use jet colormaps. Consider using perceptually uniform alternatives, as jet can introduce visual distortions that misrepresent the data [7]. A better approach would be to match the colormap to the data type—e.g., using sequential colormaps for continuous data and diverging colormaps for anomalies. The *cmocean* package [8] provides useful options.

- Label each sub-panel clearly (e.g., A, B, C, etc.).

- Specify the plotted variables, units, and relevant domain, either within the figure or in the caption. If using shorthand notations (e.g., $R$, $U$), provide a brief explanation in the caption.

- Consider whether all figures are essential to the discussion. For example, Figure 2 (climatology) and Figure 3 (anomaly pattern) depict standard visualizations that can be easily produced with xarray. While demonstrating quick visualization is useful, these figures may not add significant value to the manuscript. At a minimum, climatology and anomaly plots could be combined into a single figure to improve conciseness.

**Technical Corrections**

L47: Please check the reference.

Section 4: The title seems incomplete. Consider rephrasing the section title to something like "Application: Atlantic–Pacific Teleconnections for ENSO Prediction" to more accurately describe the scope.

L140: Please check the reference. Do you mean Rew & Davis [9]?

L225: Do you mean hot topic?

L247: phenomenon

**References**

1. Swenson, E. Continuum Power CCA: A Unified Approach for Isolating Coupled Modes. *Journal of Climate* **28,** 1016–1030. doi:10.1175/JCLI-D-14-00451.1 (2015).
2. Richman, M. B. Rotation of Principal Components. *Journal of Climatology* **6,** 293–335. doi:10.1002/joc.3370060305 (1986).
3. Hall, K. J. C. & Acharya, N. XCast: A python climate forecasting toolkit. *Frontiers in Climate* **4.** doi:10.3389/fclim.2022.953262 (2022).
4. Brady, R. X. & Spring, A. Climpred: Verification of Weather and Climate Forecasts. *Journal of Open Source Software* **6,** 2781. doi:10.21105/joss.02781 (2021).
5. Rieger, N. & Levang, S. J. xeofs: Comprehensive EOF Analysis in Python with Xarray. *Journal of Open Source Software* **9,** 6060. doi:10.21105/joss.06060 (2024).
6. Crespo, L. R., Prigent, A., Keenlyside, N., Koseki, S., Svendsen, L., Richter, I. & Sánchez-Gómez, E. Weakening of the Atlantic Niño Variability under Global Warming. *Nature Climate Change* **12,** 822–827. doi:10.1038/s41558-022-01453-y (2022).
7. Crameri, F., Shephard, G. E. & Heron, P. J. The Misuse of Colour in Science Communication. *Nature Communications* **11,** 5444. doi:10.1038/s41467-020-19160-7 (2020).
8. Thyng, K., Greene, C., Hetland, R., Zimmerle, H. & DiMarco, S. True Colors of Oceanography: Guidelines for Effective and Accurate Colormap Selection. *Oceanography* **29,** 9–13. doi:10.5670/oceanog.2016.66 (2016).
9. Rew, R. & Davis, G. NetCDF: An Interface for Scientific Data Access. *IEEE computer graphics and applications* **10,** 76–82 (1990).

---

## Author Comment (AC3)

**Spy4Cast: a Python Tool for statistical seasonal forecast based on Maximum Covariance Analysis. Review: Anonymous Referee #1**

Duran-Fonseca, Pablo and Rodriguez-Fonseca, Belen

In this response we have gone through the comments made by Anonymous Referee #1. We appreciate the effort at revising the manuscript and we have taken into account the corrections and the suggestions of the referee.

**1 General Comments**

**1. Scientific Context and Justification:** While the manuscript provides a thorough description of MCA, it does not sufficiently position MCA within the broader landscape of statistical climate forecasting methods, such as Canonical Correlation Analysis (CCA). Given that CCA maximizes correlation-often the primary metric in forecast validation-it is important to clarify why MCA was chosen over CCA and in which scenarios it is preferable or less suitable. Additionally, Spy4Cast should be contextualized within the existing Python ecosystem for climate forecasting and analysis (e.g., XCast, xeofs, climpred), which similarly leverage xarray/dask. A discussion of the conceptual and practical advantages of Spy4Cast compared to these tools-such as what Spy4Cast offers that XCast does not-would strengthen the manuscript (see Specific Comments 3 and 8).

Thanks for the comment. In the following sections we have discussed the detailed comments. We present here a list of advantages of MCA Compared with CCA and PCA:

1. **MCA Avoids Variance Masking (vs. PCA)**:

   - PCA, when dealing with multiple variables in a single matrix [6], combines variance and covariance of variables with different variability, which can obscure covarying patterns.
   - MCA focuses only on covariance, identifying 2 different matrices which covariate differently, preventing dominance by high-variance variables [1].

2. **Handles Multicollinearity Better (vs. CCA)**:

   - CCA maximizes temporal correlation but struggles when the number of grid points exceeds observations.
   - MCA avoids this issue and is better suited for climate data with multicollinearity.

3. **Better for Coupled Climate Patterns**:

   - MCA is effective in identifying coupled modes between different climate variables (e.g., SST and rainfall).
   - Useful for analyzing sources of predictability beyond just forecasting.

In summary, MCA is a robust tool for studying climate variability, particularly when analyzing relationships between multiple variables. Singularities in Canonical Correlation Analysis (CCA) arise when the covariance matrices of the input variables are **singular or nearly singular** [8, 3, 5], which is common in climate data due to:

1. **High Dimensionality**: Climate datasets often have more grid points (spatial locations) than time steps (observations), leading to **under-determined systems** where the covariance matrix is not invertible.

2. **Multicollinearity**: Strong correlations between variables (e.g., neighboring grid points in climate data) cause redundancy, making the covariance matrix nearly singular.

**How MCA Avoids This Issue:**

- Unlike CCA, MCA **does not require inverting individual covariance matrices** but focuses directly on the covariance between the two variable fields.

- This makes MCA more stable and better suited for climate studies where multicollinearity and singular covariance matrices are common.

Thus, MCA is a **practical alternative to CCA** when working with high-dimensional climate datasets, [9].

There are other python APIs with a more operational goal, to be used for producing forecast, as XCast [4], implementing a diverse set of climate forecasting tools for operational use.

Our idea is not just to produce skillful predictions but to analyze the sources of predictability. Spy4Cast is designed to be user friendly to work with multiple data sets and provide, in an easy way, the coupled modes of covariability and the one-year out crossvalidated hindcasts for climate researchers working on teleconnections.

In addition, comparing with XCast, spy4CAST works with the MCA methodology, which is not included in XCast.

We have clarified all these aspects in the new version of the paper

**2. Scalability and Computational Efficiency:** The manuscript states that Spy4Cast can handle "large" climate datasets, but its reliance on in-memory NumPy arrays for computing the cross-covariance matrix raises concerns about scalability. High-resolution or global-scale datasets could result in excessive memory usage, potentially limiting the tool's applicability to gigabyte- or terabyte-scale datasets. The authors should clarify the realistic data-size limits of Spy4Cast and discuss potential strategies for improving scalability, such as incorporating dask.array, processing data in smaller subregions, or applying prior dimensionality reduction techniques like PCA (see Specific Comment 1).

Thanks for the comment. We have clarified these aspects in the specific comments below

**3. Scope and Flexibility:** The manuscript suggests that Spy4Cast can handle "any kind of predictor," yet the current implementation appears tailored to oceanic fields with latitude-longitude-time coordinates. While the library's specialized design provides advantages, it also seems to come at the cost of flexibility-for example, through the use of custom classes such as Dataset and Region. It would be helpful to clarify whether the tool can process land-based variables, vertical levels (e.g., depth or pressure), or multiple variables simultaneously (see Specific Comments 21, 22, and 25).

We agree with the comment of the referee and have softened this statement in the new version of the manuscript. In its current version, Spy4Cast can handle any kind of 2-D predictor (lat-lon) that we want to use to assess seasonal predictability. In this way, we could use, for example, sea ice cover, snow cover, sea surface temperature, ocean heat content, or soil moisture. Nevertheless, the predictability of current seasonal forecast is mainly led by the ocean initial state, and we are using this model to test the sensitivity of different target variables to changes in the sea surface temperature variability, although we have also tested its performance to other predictors, as heat content and, whenever the field is 2-D Spy4cast works with no problem. We have modified accordingly the text. However, it is important to notice that the preprocessing of the model is in a separate module and that model could be adapted to provide predictors in 3D as these fields are not useful at seasonal time scales. Nevertheless, as the model is designed for

testing predictability at seasonal time-scales, we think that we don't need to provide 3-D fields. If we want to find coupled patterns in the 3-D dimension, we should adapt the code in the preprocessing part, that is right. In its current form, Spy4Cast can be used, also to find coupled patterns in climate data at lag 0, just to identify links between patterns to further analyze the coupling mechanisms, and also if we want to perform down scaling. For example, we could analyze the predictability of Sea Level Pressure using SST as predictor and SLP as predictand, and, later on, we could produce predictions of high resolution rainfall fields in a small region using as predictors large scale sea level pressure fields that have been predicted using Spy4cast. We further analyze these applications in the new version of the manuscript

**4. Clarity, Structure and Documentation:** Although Spy4Cast is openly licensed and includes example scripts, the documentation remains sparse, requiring users to infer key details from the source code. The manuscript refers to preprocessing steps (e.g., detrending, filtering) but does not fully explain their implementation. Additionally, some design choices-such as the use of specialized Dataset and Region classes instead of a pure xarray-based approach-should be more clearly justified, with discussion of their advantages and limitations. The structure of the manuscript would also benefit from a clearer distinction between the two validation approaches used: historical split validation and leave-one-year-out cross-validation. Finally, a more comprehensive user manual-including details on parameter options, algorithmic considerations (e.g., the use of randomized SVD), and potential pitfalls-would enhance reproducibility and usability (see Specific Comments 2,18,19,20,29,24,28, and 27).

Thanks for the comment. We might not indicate clearly in the previous version the location of the manual but the model provides an user manual including all the options indicated by the referee, and including the implementation. More details are provided in the answers to the specific comments.

**2  Specific Comments**

**Comment 1 (L7; also L48-53, 67, 69)**

The manuscript states that Spy4Cast enables large dataset manipulation," yet the internal computations appear to rely on in-memory NumPy arrays for constructing the cross-covariance matrix. To clarify the tool's practical limits, please specify:

• The maximum feasible input size (e.g. grid dimensions) before memory constraints become prohibitive.

• Whether Spy4Cast integrates with dask.array or chunked xarray computations, specifically the computation of the SVD. If not, consider explicitly stating that Spy4Cast is suited for moderate-scale datasets but may not efficiently handle very large (gigabyte-scale and beyond) datasets.

Additionally, please revisit the use of the term "scalability" (L69). While static type-checking improves code maintain-ability, it does not directly enhance the tool's ability to handle large datasets. Consider rephrasing for accuracy.

`Dataset` class has a chunks argument that gets passed to `xaray.open_dataset`. This way, dask.array is utilized for the slicing and some part of preprocessing. For MCA and filtering the arrays are "computed" and dask is no longer utilized. For this reason, it is not possible to operate on very large datasets with the current implementation. In the future it could be implemented so that dask.array is fully leveraged.

In terms of memory usage, given our limited usage of dask, the covariance matrix has to be stored in memory so we are limited to shapes around $10^4$ to $10^5$ spatial data points. As we are working mainly with 2-D data sets we could be able to work with global data with resolutions of 0.1° lat-lon, which is very common. Nevertheless, we mainly perform simulations with sliced regions so we could also work with higher resolution data in smaller regions.

We have made sure that the manuscript reflects these limitations and puts more emphasis on the main advantage of the package, that is convenience.

**Comment 2 (L13)** *... well documented for beginners and experience programmers.*

The statement about the documentation being suitable for both beginners and experienced programmers may need reconsideration. The current documentation appears quite minimal-while one notebook demonstrates the general workflow, there is little to no explanation of the individual preprocessing steps and their implications. Expanding the documentation to provide clearer descriptions of these steps would greatly enhance usability.

In terms of documentation, we have taken into consideration this comment. To provide the necessary explanations we have included a new file called `Tutorial.ipynb` in the Spy4CastManual repository. This file performs preprocessing and MCA using only xarray and shows what the abstractions Spy4Cast provides. In our opinion this makes clear what the focus of Spy4Cast is: improve convenience and provide abstractions that reduce the users' task for configuration. During research, Spy4Cast has been proven to be helpful by eliminating bureaucracy when analyzing the relations. This way research, can be focused on the results.

For documentation we have created a webpage at `https://spy4cast.readthedocs.io/`. In there, you can find detailed descriptions of the API as well as installation instructions and examples. We have made sure that the manuscript makes focus on this.

**Comment 3 (L16-40)**

The manuscript provides a detailed discussion of MCA, while Canonical Correlation Analysis (CCA) is mentioned only briefly. Since many operational forecasting systems rely on CCA-given that it maximizes correlation, which often aligns more directly with forecast evaluation criteria-please expand on the rationale for focusing on MCA.

Specifically, clarify the trade-offs between MCA and CCA in statistical forecasting. Additionally, consider citing or discussing more recent literature (e.g., [1]) that compares these methods.

If feasible, a brief note on whether Spy4Cast could be extended or adapted for CCA-or if it is specifically designed for MCA-would be valuable.

Thanks for the comment. It is true that the CCA is broadly used in statistical seasonal forecast. Nevertheless, our aim is not to provide skillful predictions but to evaluate predictability. We think that the maximization of the covariance is more interpretable than the correlation when analyzing coupled patterns in climate data. In addition, the correlation calculation can produce non-interpretable values in regions of low variability. We have experience in the use of this methodology and we did not find this methodology implemented in other tools as XCast. Nevertheless, it is not difficult to add CCA to the application, because we would just need to change the matrix to be diagonalized [9]. Wilks (2014) compares different methodologies, including CCA and MCA and they state that "Overall, the three methods exhibited generally similar skill levels"..." There was, however, a tendency for the MCA forecasts to be somewhat more skillful when trained on the full (1895 onward) predictand data record, as well as yielding better forecast calibration. MCA forecasts were decisively preferred when trained on a shorter (1951 onward) record" There are many statistical techniques that are useful when making predictions for a number of different stations or gridpoints. Canonical correlation analysis (CCA) is a statistical method creating pairs of linear combinations of the variables that have maximal correlation.

Maximum covariance analysis (MCA) is, as well, a statistical method which creates a pair of linear combinations of the variables that have maximum covariance. Rieger et al (2021) discuss the advantages of using MCA with other techniques as multivariate PCA. In this way, they state that "Climate phenomena with different expression in oceanic and atmospheric variables, such as El Niño–Southern Oscillation (ENSO), however, require the simultaneous analysis of several variables for a more comprehensive description. In principle, multivariate PCA [6] makes it possible to extract the patterns of covariability of more than one variable. However, multivariate PCA accumulates the variance and the covariance of variables with very different variability in the same quantities. In consequence this may mask covarying patterns as low-variability patterns of one variable can be erroneously accumulated in very dominant structures of one of the other, large-variability variables [1]. Maximum covariance analysis (MCA) avoids this problem masking by taking into account only the covariance between two sets of variables. As such, it bears similarity to canonical correlation analysis (CCA), which aims at maximizing the temporal correlation

between both variables. When the number of grid points (i.e., number of time series) is higher than the number of observations (i.e., number of time steps) and the data exhibit multicollinearity, as is often the case for climate data, CCA fails as it requires the individual variance matrices to be non-singular unless regularized [8]; [3] ; [5]. If the two fields of variables are identical, MCA reduces to PCA, the former thus being a natural generalization of PCA." (see also [7].

If the predictands are inter-correlated, it is possible for predictions at one or more of the locations to be somewhat inconsistent with those at others because of different sampling errors in the estimated regression coefficients, or even in the selection of predictors, for models at neighboring sites. There are various techniques that can be used to make predictions at a set of locations. These techniques include canonical correlation analysis (CCA), redundancy analysis, and maximum covariance analysis (MCA).

Covariance is a statistical parameter which indicates the direction of the linear relationship between two variables and how they vary, while correlation measures both the strength and direction of the linear relationship between two variables. It is also worth noting that covariance indicates how two variables change together, not whether one variable is dependent on another. Covariance is useful for determining the relationship; however, it is ineffective for determining the magnitude.

MCA is more useful in identifying coupled modes of, for example, SST fields and rainfall that may provide a basis for seasonal forecasting [7].

**Comment 4 (L21 onward)**

The manuscript consistently refers to one field as the "predictor" and the other as the "predictand," though MCA itself is inherently symmetric and does not imply a directional causal relationship. To avoid potential misunderstanding, consider clarifying that this terminology is used specifically in the context of forecasting rather than as a fundamental property of MCA.

Thank for this comment. We totally agree that MCA does not imply a directional causal relationship just a statistical link that maximized the covariance between two fields. We have clarified this in the paper, indicating that we use this terminology in the context of forecasting.

**Comment 5 (L21)** *In this context, MCA analysis provides spatial patterns of the predictor and predictand field which are related by teleconnections.*

The text suggests that all MCA modes inherently represent teleconnections or causal relationships. Please clarify that these modes primarily capture statistical covariance and do not necessarily indicate causal links.

Thanks for this clarification. In the same way that in the previous question, we have rewritten the text to clarify it and put in context the terminology used. MCA finds couple patterns linked by having maximum covariance, finding patterns that are coupled, and more physical analysis is needed to justify the causal relation found. It is a technique that has been broadly suggested for when analyzing impacts and drivers for teleconnections. In the context of forecast, we can use a variable that we name "predictor field" which leads in time a variable to be predicted (that we name "predictand field"). Due to the lagged relation, we can infer a hypothesis about causal links, and use this information to produce predictions. We have clarified these aspects in the corrected version.

**Comment 6 (L33)** *... it has the advantage of being easily interpretable ...*

Higher-order MCA modes can become increasingly difficult to interpret due to the orthogonality constraints of SVD, potentially leading to so-called "Buell patterns" (similar to PCA) [e.g., 2]. Please acknowledge this limitation when discussing the interpretation of the second or third MCA modes.

Thanks for this clarification. We have added this limitation in the text. In most of the applications that we have done of this methodology we have just used the first mode. Nevertheless, for some teleconnections, the second and third modes have been found to be interpretable and add skill when including these modes in the calculation of the skill. We include here 2 cases:

**Case 1:** Sahel rainfall-Pacific SSTs for which the addition of more modes do not add information figure: 1.

[Figure]

Figure 1: Top: coupled modes of variability identified using Spy4CAST for anomalous JAS SST in the Pacific and anomalous JAS rainfall in the West African region. Bottom: anomaly correlation coefficient (ACC) calculated using a leave-out-one year crossvalidation technique in which in the first column just one mode is included, in the second 2 modes are included and in the third 3 modes are included. The reconstruction of the rainfall data is skillful if we just consider one month. The inclusion of more modes do not improve the results.

**Case 2:** European SLP impact on rainfall in spring. We know that European rainfall variability can be led by different modes of variability affecting differently (North Atlantic Oscillation, East Atlantic pattern etc). The identification of these coupled modes and their inclusion in the prediction can improve the predictability 2.

**Comment 7 (L41)** *... a new paradigm of research in climate variability studies has emerged ...*

This statement is vague for readers unfamiliar with Cai et al. (2019). Please specify what this paradigm entails (e.g., cross-basin interactions, trans-basin teleconnections) and clarify how Spy4Cast relates to it.

When using Spy4CAST for illustrating an example, we decided to apply it for the case of tropical basin interactions, as this is a new paradigm of research in which Pacific El Niño (La Niña) has been found to be predictable in certain decades, with 6 months in advance. The predictor found is the equatorial Atlantic. This tropical basin interactions and others are described in [2]. Spy4cast can be used to assess this predictability by using the tropical Atlantic SST anomalies in JAS as "predictor field" and the tropical Pacific SST anomalies as "predictand field" corroborating that the potential predictability of Pacific El Niño (La Niña) based on the Atlantic SSTs.

[Figure]

Figure 2: Top: coupled modes of variability identified using Spy4CAST with SLP and rainfall in FMA . Bottom: anomaly correlation coefficient (ACC) calculated using a leave-out-one year crossvalidation technique in which in the first column just one mode is included, in the second 2 modes are included and, in the third, 3 modes are included The reconstruction of the rainfall data is skillful if we consider more than the first modes.

**Comment 8 (L48-53)**

The manuscript would benefit from a more thorough discussion of other Python-based forecasting and dimensionality-reduction packages, such as XCast [3], climpred [4], and xeofs [5]. Please compare Spy4Cast's approach, strengths, and limitations relative to these tools-for example, in-memory vs. distributed computing or a specialized feature set vs. a more general framework. Expanding this discussion would help clarify Spy4Cast's positioning within the broader climate data analysis software ecosystem.

We have included in the new version the advantages of the use of Spy4CAST and MCA compared with other packages

In general these are the advantages

1. Climpred is used to analyse the skill produced by operational seasonal forecasts but do not produce seasonal forecast and do not identify coupled modes of variability in climate data. It is designed to evaluate predictions.

2. Spy4cast just need the files and a few parameters to produce coupled patterns and crossvalidated hindcast together with skill parameters, as for example when relating SST and rainfall providing a basis for seasonal forecasting [7]. All preprocessing and data management is included in Spy4cast (not in xEOF and more complicated in Xcast)

3. Spy4cast uses MCA which is not implemented in Xcast.

4. Xcast has a more operational goal, to be used for producing forecast [4], implementing a diverse set of climate forecasting tools for operational use. Spy4cast is not designed to produce skillful predictions but to assess sources of predictability and coupled patterns in climate data. Spy4CAST is designed to be user friendly to work with different types of data sets and provide, in an easy way, these coupled modes of covariability and the one-year out crossvalidated hindcasts for climate researchers working on teleconnections.

**Comment 9 (L83)**

The definition of the cross-covariance matrix assumes zero-mean time series. Additionally, the formulation appears to be missing a normalization factor of $1/n_t$ or $1/(n_t - 1)$ for an unbiased covariance estimation. Please clarify or correct this as needed.

Thanks for the clarification. It has been corrected

**Comment 10 (L89)** *The information of this matrix is redundant ...*

It would be more precise to describe this as a high degree of redundancy-or even better, as multicollinearity.

Thanks for the clarification. It has been rewritten following this suggestion.

**Comment 11 (L89)** *... produce the same maps ...*

This statement is generally incorrect. You likely mean "similar maps" rather than identical ones. The claim would only be true if two time series were exactly equal, which is unlikely in practice.

Thanks for the clarification. We agree that this statement is not correct and has been corrected accordingly.

**Comment 12 (L89)** *Also it is a complex matrix as it takes into account all possible relations between points.*

Are you certain this includes *all* possible relationships, or only linear ones? Consider revising for accuracy. Additionally, rather than making this statement, it may be more useful to emphasize that the matrix grows rapidly in size, with dimensions *nyxnz*.

Thanks for the suggestion. We have added the fact that all possible "linear" relations are included and also that the matrix grows rapidly in size.

**Comment 13 (L94)** *... matrix of eigenvalues ...*

The result of SVD provides a diagonal matrix containing the singular values, not eigenvalues. Please correct this terminology.

We have changed the name accordingly.

**Comment 14 (L94)** *... which represents the squared covariance fraction ...*

Each singular value represents the covariance explained, while the squared singular values correspond to the squared covariance. The Squared Covariance Fraction (SCF) can be computed from the singular values, but it is not provided directly.

We infer the SCF from the singular values. We have pointed out this fact to be more precise.

**Comment 15 (L100)** *... fraction of variance ...*

fraction of squared covariance

We infer the SCF from the singular values. We have pointed out this fact to be more precise.

**Comment 16 (L101)** *... which are linked by having maximum covariance.*

More precisely, the first mode is linked by maximum covariance. Subsequent modes follow these principles: (i) Mode $i + 1$ captures the maximum covariance of the remaining data after the first $i$ modes have been removed. (ii) This is subject to the constraint that the patterns $R_{i+1}$ and $Q^{i+1}$ are orthogonal.

Thanks for this suggested text. We have added it in this new version "More precisely, the first mode is linked by maximum covariance. Subsequent modes follow these principles: (i) Mode $i + 1$ captures the maximum covariance of the remaining data after the first $i$ modes have been removed. (ii) This is subject to the constraint that the patterns $R_{i+1}$ and $Q^{i+1}$ are orthogonal."

**Comment 17 (L120)** *... where n is the number of observations.*

I assume that $n$ refers to the number of grid points or spatial locations. The term "observations" may imply a temporal scale, which does not seem to be the case here. Please clarify.

We really appreciate all these inputs. We have changed "observations" by "number of grid points"

**Comment 18 (L134-135)** *Spy4Cast is organized in three steps: setup, preprocess and methodology. The procedural workflow is illustrated in figure 1.*

The text appears to contradict Figure 1, which explicitly separates configuration and methodology, while treating preprocessing as part of methodology (see Specific Comment 46). Please clarify or revise for consistency.

We have changed this figure to address this comment and the other one relating to Figure 1. In the revised manuscript we have included figure 3.

[Figure]

Figure 3: New workflow diagram

**Comment 19 (L135-136)**

Figure 1 presents a logical workflow, yet this structure does not appear to be reflected in the design of the software. Could you clarify the reasoning behind this discrepancy?

The new workflow diagram attempts to solve this issue.

**Comment 20 (L139)** *Configuration: loading data and slicing region.*

I would argue that loading data and slicing a region are distinct from configuration. Shouldn't configuration be limited to defining metadata, with data loading and preprocessing occurring separately based on that configuration? Merging these logically different concepts seems confusing. Could you clarify the

reasoning behind this design choice?

Thanks for the comment. We have rethought the name "configuration". We have made it clearer in the revised manuscript.

**Comment 21 (L140-165)**

It seems that some functionalities, such as temporal and spatial slicing, are already available in xarray. Could you clarify why you chose to introduce custom classes like Dataset and Region instead of relying on xarray's native methods?

Additionally, does Spy4Cast support advanced xarray features like openmfdataset, which are essential for working with climate datasets distributed across multiple files?

Finally, for users who need custom preprocessing steps (e.g., weighting, detrending), can they seamlessly revert to or integrate with standard xarray workflows?

The reason why we introduce these classes is to remove the users' need to know how to slice properly the dataset. In some cases, datasets may use different conventions and your code will be different for each dataset. Also, this API allows unexperienced users to slice a dataset easily. In the new tutorial `https://github.com/pabloduran016/Spy4CastManual/blob/main/Tutorial.ipynb` we show how much easier is to use Spy4Cast than xarray to slice the array.

In the current implementation we don't work with `open_mfdataset`. We have not needed it, even though we have tested this API in big datasets (we have run Spy4Cast on CMIP6 datasets). If anyone needs that functionality we would be glad to modify the code accordingly.

To implement custom preprocessing steps you can create a subclass of `Preprocess`. We have created a new example in the repository to show it `https://github.com/pabloduran016/Spy4Cast/blob/main/examples/how_to_use_custom_preprocessing_steps.py`.

**Comment 22 (L140-165)**

Currently, Spy4Cast appears to support only 2D or 3D ocean-centric data (longitude, latitude, time). If the intention is to allow "any kind of predictor" (L9,10), please clarify whether different variables (e.g., global soil moisture) or multi-dimensional data (e.g., depth, forecast members) can be seamlessly incorporated.

Additionally, do the constraints for the predictor apply equally to the predictand?

If the current implementation is specialized (e.g., limited to ocean-only fields or requiring specific naming conventions), please make these limitations explicit in the user documentation.

Thanks for this comment. As we clarified previously, Spy4Cast, in its present form, is being used to work with 2D variables but not necessarily oceanic ones. They can be also sea ice cover, snow cover, soil moisture or any other variable that can drive the variability of a target climate variable. Also, we can work in lag 0 to identify coupled patterns in climate data, without doing predictions (just reconstructions, for example, to do downscaling), using, for example sea level pressure and rainfall, as Y and Z variables respectively. Spy4Cast is designed also to work with different forecast times and start dates and can be used to compare operational seasonal forecast with a prediction done with Spy4Cast.

We present here an example, in Fig2, in which we compare the skill of the SEAS5 predictions (seasonal hindcasts produced by the European Center for Medium Range Weather Forecast, ECMWF) and those produced by Spy4CAST. We can see the power of Spy4CAST in the ACC maps, as we compare the skill produced by both predictions. Oerational seasonal prediction systems (in the example done we test it for the prediction of Nov-Dec sea level pressure at global scale using, as predictor field, the October SSTs). In addition, we have recently coupled the Spy4CAST to the ESMValtool to analyze predictability of European rainfall in future scenarios compared with historical runs. We want to highlight that power of the tool is its simplicity as it just need some parameters to be run. The user just needs to indicate

[Figure]

[Figure]

Figure 4: Forecast skill, expressed in terms of the anomaly correlation coefficient (ACC) between (top) anomalies in ERA-5 sea level pressure in November-December predicted using a one-year out crossvalidated hindcast with Spy4Cast using as predictor the anomalies in HadISST SST in October. (bottom) SEAS5 ensemble mean in SLP in early-winter (Nov-Dec) using the initialization of October. Only areas where the skill is significant at the 95% confidence level are represented

the location of 2 data sets, the name of the files, the regions for the 2 variables to be linked, the seasons to be considered in the relation, the cut off frequency for the filter, the start date, the forecast time and significance level of the results.

**Comment 23 (L144)** *This method will not load the data-set into memory, because it internally uses xarray function xarray.opendataset*

xarray.opendataset does not inherently prevent loading the dataset into memory. The key factor is whether the chunks argument is specified. Please clarify this point.

I don't think I understand this comment. In my opinion, `open_dataset` won't load the dataset into memory. By using `open_dataset`, the dataset can be sliced before loading it into memory and the speed can be noticed in the code when using it (without passing any `chunks` argument).

According to xarray documentation (`https://docs.xarray.dev/en/stable/generated/xarray.open_dataset.html`):

`open_dataset` opens the file with read-only access.

As opposed to `load_dataset` (`https://docs.xarray.dev/en/stable/generated/xarray.load_dataset.html`):

Open, load into memory, and close a Dataset from a file or file-like object. It differs from `open_dataset` in that it loads the Dataset into memory, closes the file, and returns the Dataset. In contrast, `open_dataset` keeps the file handle open and lazy loads its contents. All parameters are passed directly to `open_dataset`. See that documentation for further details.

**Comment 24 (L168-179)** *This preproccesing includes calculation of seasonal means, computation of seasonal anoma-lies and filtering. [... ] Next, the MCA is applied, with a time linear-detrending of the data by default.*

The methods for time filtering and linear detrending require more detail:

• Is the linear detrending applied gridpoint-wise, or across the monthly or seasonal time series?

• What does "frequency filtering" entail (e.g., Butterworth filter parameters, cutoff frequencies)?

• How is missing data handled (e.g., masked land areas)?

Consider including these details in the "Preprocessing" section for clarity.

Additionally, if Spy4Cast is primarily designed for ocean variables, please specify how continental points and other 3D dimensions (e.g., depth or altitude) are treated.

We have made sure the revised manuscript clarifies these points. In particular, the linear detrending is done for the time series of each particular grid points. The filter is a high pass butterworth filter, in which you can choose, as inputs, the order and the period (cut off frequency). The masked areas are not included in the computation. For oceanic variables, the land points are not used in the calculation.

**Comment 25 (L172)** *All the data can be stored in .npy format.*

Storing preprocessed data in .npy format may reduce compatibility with xarray/dask workflows and is less common for data exchange within the climate science community. Could you clarify the rationale for choosing .npy over standard formats like netCDF or Zarr, which are widely used for large-scale climate data? If this choice is primarily for internal convenience, please state that explicitly.

The choice of `.npy` format is due to internal convenience in the sense that it is a easy to use format. You just need to understand two functions: `np.save` and `np.load`. The output of `np.load` is the same as using the corresponding attribute of the objects created (`MCA, Crossvalidation, ...`).

It is true that it can be convenient to use other formats. If a user wants to do so, they can do it in a few lines of code.

With our workflows we found it worked fine for us, but if we find that a number of users request that functionality, we can implement it.

**Comment 26 (L187)** *MCA and Crossvalidation*

It would be clearer to separate MCA and cross-validation into two distinct sections. In the MCA section, please specify that you use approximate algorithms based on randomized linear algebra to accelerate singular value decomposition and provide relevant references.

Thanks for the comment. We have separated both methodologies in separate sections and, also, clarify the explanation in the revised manuscript, providing relevant references.

**Comment 27 (L190)** *MCA uses test-t significance technique ...*

How is the t-test applied-one-sided or two-sided? What is the target variable? I assume it is the correlation coefficients of the homogeneous/heterogeneous correlation patterns. Additionally, please account for multiple testing when interpreting p-values or discuss how this issue is addressed.

Thanks for the comment. T-test is used for assessing significance of the correlation coefficients either as

an score or in each grid point when calculating the homogeneous and heterogeneous correlation map. It is a two-tailed test. This has been clarified in the new version.

**Comment 28 (196)** *... can be used to calculate other regression maps using different variables and datasets for the same period analysed (see mca and indexregression in Duran-Fonseca and Rodriguez-Fonseca (2024b)).*

Please clarify this statement in more detail. Referring readers to the mca and indexregression submodules of Spy4Cast without further explanation makes it difficult to understand the intended meaning without digging through the code. Providing a brief description of how these regression maps are computed would improve clarity.

Thanks for the comment. We have modified this in the new version of the manuscript, indicating the way the regression maps are calculated.

**Comment 29 (L211)** *Arguments mapy and mapz were used to create a global regression along a larger region.*

Please elaborate on the meaning of mapy and mapz. What exactly do these arguments represent? Additionally, the phrase "global regression along a larger region" is unclear-a larger region is not necessarily global. Could you clarify what is being regressed against what?

Thanks for the comment. This calculation is done for plotting global maps for the regression, in stead of plotting the regression for the region used in the MCA. We have modified the manuscript in this regard.

**Comment 30 (L205-225)**

The manuscript would benefit from a more structured explanation of validation and cross-validation:

• I suggest consolidating both discussions into a single section, as they are closely related. While cross-validation employs classical leave-one-out cross-validation (LOO-CV), the validation approach instead partitions the dataset into concurrent (historical) time slices to account for long-term temporal autocorrelation.

• Additionally, please clarify the apparent contradiction between "Spy4Cast is not designed to assess stationarity"(L56) and "Spy4Cast is able to [...] look for non-stationary relations" (L215). Clearly defining the tool's actual capabilities and limitations in this context would improve consistency.

Thanks for the comment. There is no contradiction, although we have changed the text to avoid miss-interpretations. The tool can be used for performing sensitivity experiments analyzing the different modes of co-variability changing the period. In this way, it can be used for analyzing non stationary relations by comparing the results obtained in different periods. Nevertheless, it cannot be used for analyzing non stationarities directly,in a single realization, as it is necessary to perform different analysis for that.

**Comment 31 (L226,227)**: *Using spy4cast, sensitivity experiments could be defined, training with different periods to identify those validating periods which follow the same modes of covariability*

I find this sentence difficult to understand (apologies, as I am not a native English speaker). Could you rephrase it to clarify the intended meaning?

Thanks for the comment. The sentence has been rephrase as: "Using spy4cast, sensitivity experiments could be defined, training with different periods and comparing the modes of covariability to identify the periods that identify the same modes. For example, Tropical Atlantic can be related or not with anomalies in the Pacific. Thus,by performing MCAs in different training periods we can identify those periods following this teleconnection. ".

**Comment 32 (L229-240)**

To provide broader context and strengthen the demonstration of Spy4Cast's applicability, consider incorporating recent references on Atlantic Niño weakening under climate change (e.g., Crespo *et al.* [6]). This would further highlight the tool's relevance to current climate research questions.

Thank you very much for this suggestion. We have included the references suggested. In addition, we are currently using this tool for analyzing CMIP6 experiments under different climate change scenarios, so it could be used, for sure, to better understand changes in Atlantic Niño impacts in recent decades.

**Comment 33 (L241)** *... all these features are well represented by spy4cast.*

Could you clarify what is meant by this statement? For example, earlier, you mention that the Atlantic phenomenon peaks in boreal summer, but this appears to be an a priori choice in the modeling process rather than an outcome inherently produced by the software.

Do you mean that these features can be accommodated by the researcher when using Spy4Cast, implying that the key strength is its flexibility? If so, please rephrase for clarity.

Thanks for the comment. We have clarified this fact. We mean that the results found by other authors in which the boreal summer Atlantic Niño was found to be a predictor of La Niña (and viceersa for Alantic La Niña) is validated using this methodology (applying Spy4CAST)

**Comment 34 (L244)** *... different variables are created.*

Please specify which variables are created. Are these evaluation metrics, MCA results, or something else? Figure 4 only displays file names, making it difficult to determine this with certainty.

Thanks for the comment. We have described the variables accordingly in the revised manuscript.

**Comment 35 (L244)** *Spy4cast identifies the Atlantic Niño in JJA as the main mode of covariability with DJF Pacific SST anomalies.*

While Spy4Cast does identify the Atlantic Niño in JJA as the main mode of covariability with DJF Pacific SST anomalies, this result is largely determined by the specific choice of seasons and regions (tropical Pacific and tropical Atlantic) in the analysis. Given these constraints, what alternative outcomes could have emerged? Would the result change if the predictor region were expanded?

In the introduction, you mention that MCA can help guide more advanced prediction algorithms by identifying potential predictor regions. However, your case study presupposes prior knowledge of the predictor region. Would it be more aligned with your introduction to assume little to no a priori knowledge and let MCA reveal potential predictors (e.g., Atlantic El Niño)?

This is not necessarily a suggestion for the revised manuscript, but rather a potential idea for future exploration.

Thanks for this comment. It is true that this method is more useful if the user knows a priori some connections found and lags. Nevertheless, we can start an analysis in lag 0 and repeat the MCA and the crossvalidation with different lags in order to find predictability patterns. We have clarified these aspects in the new version of the paper

**Comment 36 (L246-247)** *This expansion coefficient is highly correlated with all grid points in the equatorial Atlantic, shaping the Atlantic Niño phenomenon.*

For example, you could compute the Pearson Please provide a quantitative measure to support this statement.

correlation coefficient between the expansion coefficients and the Oceanic Niño Index (ONI) to quantify their similarity to ENSO.

Yes, one can correlate the expansion coefficients with the ONI but, I do not think that it is necessary as the SUY regression map spatial configuration is telling us that we have found a Pacific El Niño , giving even more information than the ONI. For example, using MCA different El Niño flavor could emerge as

separate modes, for example, Eastern Pacific (EP) and Central Pacific (CP) Niños (Niñas).

**Comment 37 (L247)** *This leading mode explains almost the 60 % of the covariability*

60 % of the **squared** covariance.

Thanks. It has been corrected

**Comment 38 (L249)**

**squared** covariance

Done

**Comment 39 (L257-258)** *This fact does not hold for the whole period, suggesting that the linear nature of this methodology cannot always produce accurate predictions, as there are other non-linear relationships that the MCA is not able to capture.*

You could note that a discrepancy between different evaluation metrics (e.g., high ACC but high RMSE) often indicates a bias in the error distribution. This bias may stem from the inherent linear assumptions of MCA, which, as you correctly point out, can limit its ability to predict nonlinear extreme events.

Thanks for the clarification. We have noted this interesting fact in the text.

**Comment 40 (L263)** *This API [... ] has proven effective in increasing productivity and the quality of research.*

Unless there is concrete evidence supporting this claim, I would suggest softening the statement. Instead, you could say that the API has the potential to improve productivity and reproducibility in research.

You are right. The statement was too strong. We have changed it following your suggestion by: "this API has the potential to improve productivity and reproducibility in research."

**Comment 41 (L265)** *Spy4Cast represent the beginning of a new approach to statistical seasonal forecasting ...*

What exactly do you mean by "new approach"? MCA itself is not new-are you referring to the use of predefined routines for analysis? If so, this is also not entirely novel, as operational forecast centers routinely employ such methods for statistical seasonal forecasting. To ensure accuracy, consider avoiding broad claims about "new approaches" unless specific evidence or metrics are provided.

Sorry about this statement. With new approach we wanted to stress the fact that Spy4cast could be used as a tool to test different data sets in an easy and rapid way, in order to make assessment of seasonal predictability. But, we totally agree, reading again the sentence, and we have omitted this sentence.

**Comment 42 (L267)** *... this API is more versatile ...*

Please clarify what makes this API "more versatile" compared to existing open-source solutions. Providing specific examples or comparisons would help substantiate this claim.

versatil means "Able to adapt easily and quickly to various functions" Although there are other APIS this is faster and very easy to install and use.

**Comment 43 (L270)** *Indeed, within the OFF project, it is being integrated into ESMValTool.*

Could you provide more details on the planned integration with ESMValTool, such as the expected timeline and intended functionality? Additionally, if "OFF" refers to a specific project, please define the acronym for clarity.

We have added more details about the functionality of integrating ESMValTool with Spy4CAST. OFF refers to a spanish project "Oceans for Future: Integrating tools for .....". We have indicated it in the

aknowledgements and in the text.

**Comment 44 (Conclusion and Discussion)**

Consider separating the discussion of the tool's limitations and future directions from the concluding remarks to follow a more standard "Conclusion and Outlook" structure.

We have done this separation in the new version of the manuscript.

**Comment 45 (Listings 1-10)**

The numerous short code listings may quickly become outdated if the API changes. Consider moving them to an online supplement or user guide while keeping only a concise set of essential examples in the main text. This would help maintain the paper's focus while ensuring comprehensive examples remain accessible in the documentation. Additionally, reassess whether all listings are necessary in their current form-Listing 2, for instance, provides limited information.

Thanks for the comment. We have taken it into account and removed the listings from the revised version of the manuscript. We agree it improves readability.

**Comment 46 (Figure 1)** Could you clarify the meaning of the left-hand-side arrow? Additionally, there appears to be an inconsistency between the manuscript (L134) and Figure 1 regarding the workflow structure. The text describes three steps: configuration, preprocessing, and methodology (which includes MCA and validation). However, in the figure, the workflow is grouped into only two categories: configuration and methodology, with preprocessing included under methodology. I would argue that temporal and spatial slicing operations are also part of preprocessing.

Please explain and justify why the workflow is structured this way. Additionally, the different abstraction levels implied by the colors, boxes, and shapes are somewhat unclear. A more explicit explanation of how these visual elements correspond to the workflow's logical structure would improve clarity.

We have changed the figure refering to the structure of the API.

**Comment 47 (Figure 4)**

This figure is difficult to interpret without additional context. If it is meant as a quick reference, consider adding a brief explanation in the caption about the typical use and content of each array. Otherwise, reassess whether the figure is essential for the manuscript's long-term clarity and sustainability (cf. Specific Comment 45).

It is true that it needs more context, we have provided it.

**Comment 48 (Figure 5)**

It appears that this figure presents the output of the MCA (expansion coefficients and homogeneous/heterogeneous correlation patterns for modes 1 to 3), yet the caption states "predicting Niño." However, no actual prediction is shown-only the covarying patterns of variability between time-lagged SST in the tropical Pacific and Atlantic.

The idea is to predict El Niño, so that is the reason of indicating that in the figure. It is true that, a priori we do not know what is going to be the result,so this is not correct. We have changed the title by "Figure 5. Fast plot of MCA outputs. Spy4CAST is applied for sea surface temperature of the equatorial Atlantic and Pacific oceans, with a lag of 7 months between the Atlantic and the Pacific (leading the Atlantic).modes 1 to 3 are represented. Period: 1977 to 2001."

**Comment 49 (Figure 6)**

When presenting ACC and RMSE, please clarify the temporal dimension over which they are computed and specify the reference variable or index. Additionally, provide units for RMSE (presumably ∘C).

It would also be helpful to briefly discuss the error in the context of SST variability-is the RMSE relatively low or large compared to typical SST variations?

Thanks for the comment. In the new version, we have clarified the temporal dimensions and provide the units. In general, the RMSE obtained by this methodology is big, because the methodology maximizes the variance between the expansion coefficients, prioritizing this and not minimizing the error. Thus, the ACC is high but also the RMSE. In this new version, we have briefly discussed these aspects.

**Comment 50 (Figure 6)**

Please provide a more detailed description in the caption. Specifically, what do the orange dots in the upper-right panel represent? Do they indicate uncertainties in ACC values? If so, how are these uncertainties calculated?

Thanks for the comment. Orange dots indicate the times where the correlation is significative, $p_{val} < \alpha$. The caption of the equivalent figure (now figure 4) is: " Results from the Crossvalidated hindcast produced for the MCA between Pacific SST anomalies in DJF and Atlantic SST anomalies in the previous JJA. Top panel represent the spacial skill as anomaly correlation coefficient (ACC), with black dots indicating the regions which are significance. Right figure on the top represents the ACC for each of the years, as a result of correlating the observed and crossvalidated maps in each time step. Indicated with orange dots, the years where the correlation is significative, $p_{val} < \alpha$. In the bottom, the left figure represents the maps of Root Mean Squared Error (units in units of $Z$, $°C$) between the observed SSTs and the crossvalidated hindcast. The right figure in the bottom represents the RMSE for each of the years, as a result of calculating this RMSE between the observed and crossvalidated maps in each time step. Period: 1977 to 2001".

**Comment 51 (Figure 7)**

RMSE is typically non-negative. Could you clarify how a negative RMSE appears in the figure?

Thanks for the comment. There was an error that has been solved in this new version.

**Comment 52 (Figures-General)**

• Some figures (particularly Figures 2, 3, 6, and 7) appear to use jet colormaps. Consider using perceptually uniform alternatives, as jet can introduce visual distortions that misrepresent the data [7]. A better approach would be to match the colormap to the data type-e.g., using sequential colormaps for continuous data and diverging colormaps for anomalies. The *cmocean* package [8] provides useful options.

We have changed the figure referring to the structure of the API.

• Label each sub-panel clearly (e.g., A, B, C, etc.).

In the new version we have labeled each of the sub-pannels. Thanks for the suggestion.

• Specify the plotted variables, units, and relevant domain, either within the figure or in the caption. If using shorthand notations (e.g., $R$, $U$), provide a brief explanation in the caption.

Thanks for the comment. In this new version of the manuscript we have followed all these suggestions when plotting variables and better explained figure captions

• Consider whether all figures are essential to the discussion. For example, Figure 2 (climatology) and Figure 3 (anomaly pattern) depict standard visualizations that can be easily produced with xarray. While demonstrating quick visualization is useful, these figures may not add significant value to the manuscript. At a minimum, climatology and anomaly plots could be combined into a single figure to improve conciseness.

We took this into account and combined the two figures into one.

**Technical Corrections**

L47: Please check the reference.

Thanks for the comment. Sorry for not including that reference. The reference was: Counillon, F., Keenlyside, N., Toniazzo, T., Koseki, S., Demissie, T., Bethke, I., & Wang, Y. (2021). Relating model bias and prediction skill in the equatorial Atlantic. Climate Dynamics, 56, 2617-2630.

Section 4: The title seems incomplete. Consider rephrasing the section title to something like "Application: Atlantic-Pacific Teleconnections for ENSO Prediction" to more accurately describe the scope.

thanks for the correction. It is true that the title was not complete. The title of the section is : Application to the study of the Equatorial Atlantic impact on the tropical Pacific

L140: Please check the reference. Do you mean Rew & Davis [9]? Yes we did. We have corrected it

L225: Do you mean hot topic? Yes, we have changed to "hot topic" in the new version

L247: phenomenon Thanks for the comment. We have changed it accordingly

**References**

[1] Christopher S Bretherton, Catherine Smith, and John M Wallace. An intercomparison of methods for finding coupled patterns in climate data. *Journal of climate*, 5(6):541–560, 1992.

[2] Wenju Cai, Lixin Wu, Matthieu Lengaigne, Tim Li, Shayne Mcgregor, J.-S Kug, Jin-Yi Yu, Malte Stuecker, Agus Santoso, Xichen Li, Yoo-Geun Ham, Yoshimitsu Chikamoto, Benjamin Ng, Michael McPhaden, Yan Du, Dietmar Dommenget, Fan Jia, Jules Kajtar, Noel Keenlyside, and Ping Chang. Pantropical climate interactions. *Science*, 363:eaav4236, 03 2019.

[3] Raul Cruz-Cano and Mei-Ling Ting Lee. Fast regularized canonical correlation analysis. *Computational Statistics & Data Analysis*, 70:88–100, 2014.

[4] Kyle Joseph Chen Hall and Nachiketa Acharya. Xcast: A python climate forecasting toolkit. *Frontiers in Climate*, 4:953262, 2022.

[5] Abdel Hannachi. Regularised empirical orthogonal functions. *Tellus A: Dynamic Meteorology and Oceanography*, 68(1):31723, 2016.

[6] John E Kutzbach. Empirical eigenvectors of sea-level pressure, surface temperature and precipitation complexes over north america. *Journal of Applied Meteorology and Climatology*, 6(5):791–802, 1967.

[7] Simon J Mason and Omar Baddour. Statistical modelling. In *Seasonal climate: forecasting and managing risk*, pages 163–201. Springer, 2008.

[8] Hrishikesh D Vinod. Canonical ridge and econometrics of joint production. *Journal of econometrics*, 4(2):147–166, 1976.

[9] Daniel S Wilks. Probabilistic canonical correlation analysis forecasts, with application to tropical pacific sea-surface temperatures. *International journal of climatology*, 34(5), 2014.

---

## Author Comment (AC4)

**Spy4Cast: a Python Tool for statistical seasonal forecast based on Maximum Covariance Analysis. Review: Anonymous Referee #2**

Duran-Fonseca, Pablo and Rodriguez-Fonseca, Belen

In this response we have gone through the comments made by Anonymous Referee #2. We appreciate the effort at revising the manuscript and we have taken into account the corrections and the suggestions of the referee.

**1 Major Comments**

Given the current scope of the software, I strongly recommend providing thorough documentation and a user manual. Additional examples of its application would further enhance its value to the community.

Thank you for your comment. Anonymous referee #1 had a similar comment regarding documenation. To adress it we made a clearer reference to the documentation (currently changed to readthedocs.io: https://spy4cast.readthedocs.io). We have also improved the user manual at https://github.com/pabloduran016/Spy4CastManual by adding Tutorial.ipynb and other examples you are welcome to explore, which shows in detail part of the functionality that Spy4Cast implements. In the new published version, Spy4Cast repository has a folder examples which contains itself the user manual.

Furthermore, as a comprehensive tool for statistical seasonal forecasting, the software would benefit from referencing influential prior work that incorporates statistical analysis and cross-validation. This would provide a more complete methodological context. A quick Google search yields references such as 'Cross-Validation in Statistical Climate Forecast Models' by Barnston and van den Dool (1987). I would encourage the authors to conduct a more thorough search to identify additional relevant studies.

Thanks for the comment. We have added a more methodological context, including references as the one suggested (which in fact is Michaelsen [1987]), but also others in which these methodologies have been applied. An extended description of the methodology is also included. In this list of references, we include the work of Wilks [2014] in which the author compares different methodologies including CCA and MCA, suggesting the use of MCA. Also we cite Michaelsen [1987], Elsner and Schmertmann [1994], Johansson et al. [1998]

**Since this software provides a ready-to-use implementation of an existing method rather than introducing a new approach, what are its future prospects? Do you have plans to expand its functionality or broaden its applicability?**

Thanks for the comment. Spy4Cast offers a new a approach regarding its implementation, not the methodology. It is true that the way it was written in the manuscript could lead to misunderstandings and sounded pretentious. We have indicated that in the revised manuscript.

The idea behind the creation of this API was to apply it to a broad number of datasets to asses predictability. We are in the era of DATA and we can test many data sets to assess predictability of different target variables. This let us to better explore teleconnections and impacts and build hypothesis. Thus, we need a easy to run methodology that allows us to perform simulations in a fast and easy way, including the selection of regions and seasonas, the preprocessing , application of the MCA, crossvalidation and plotting. There are many applications that can be done, that we have indicated in the new version (with a link to some examples available in the manual). For example:

- 1. Identify coupled patterns in climate data (all examples)
- 2. Evaluate the ACC as a function of the lag betweeb the predictor and predictand field using different start dates and forecast times (Analysis\_of\_Predictability.py)
- 3. Assess stationarity of the teleconnections (Pacific\_Impact\_European\_Rainfall.ipynb).

In addition, other applications could be the assessments of the role of different drivers on the skill of a target variable to be predicted, the comparison between the skill of different simulations provided by different CMIP models, the comparison with the skill of dynamical forecasts etc.

In the new version of the manuscript, we have included these different applications of the MCA

**2 Detailed comments**

**Line 1: Did you mean a dimension reduction technique?**

We have changed the sentence to "Maximum Covariance Analysis (MCA) is a reduction dimension technique"

**Line 10: How do you test model sensitivity to particular years? Did you say this in the manuscript?**

Spy4Cast analyses the model sensitivity to particular years, including a diagnosis of the stability of the obtained modes to particular outliers. When we do a one-year out crossvalidation, we are repeating the MCA without the target year. In this way, we can storage the information of the mode, and test the results (scf, ruv, U, V, RUY, RUZ etc) without that year. This let us know this sensitivity. We have further explained this in the new version. An example of this is jupyter notebook Pacific\_Impact\_Sahelian\_Rainfall.ipynb where we use version=2 to plot crossvalidation (section Analyze the importance of the first 3 modes)

**Line 11: How would you test modes to particular outliers? Did you mention this? Or is this implied by testing different batch of years?**

This is explained in the previous answer and we have further explained this in the new version of the manuscript

**Line 14: Is the software fully documented? As it stands, it would benefit from additional work to develop a more comprehensive manual.**

The software is fully documented. It also has a manual and a list of examples in the source directory. It also has a small suite of unit tests to ensure safe scalability.

Line 20-25: SST and SLP patterns which are highly 'correlated'. You need to clarify an important distinction. Your description of the coupling between SST and SLP refers to only one phase of the Southern Oscillation or ENSO. However, it applies to both El Niño and La Niña, not just El Niño. The current wording suggests that only El Niño is being considered.

Thanks for the comment. In the new version of the manuscript we have changed it to: "For one of the phases of the expansion coefficients, the SST pattern will have the structure of El Niño and the SLP pattern will have the structure of the Southern Oscillation in its negative phase; which is the co-varying

SLP pattern forced by El Niño. The opposite will be obtained for the other phase of the expansion coefficients"

**Line 28: a baseline for seasonal forecasts? Why**

We want to apologize for not having further developed this idea in the previous manuscript. As dynamical models suffer for having biases, this statistical tool can serve to identify the dominant patterns of variability in the observations and help to correct or select members in operational predictions. In this way, one could combine the use of the statistical forecast provided by Spy4Cast and those provided for the different members of the dynamical seasonal forecast, calculating the ensemble mean of those members correlated with the statistical prediction.

Line 30-33: Machine learning methods are rapidly evolving, and you should reference more recent studies to support this statement. For example, Toride et al. (2024, https://arxiv.org/abs/2404.15419) demonstrates the use of neural networks to identify physical relationships and find predictability.

In this new version we have discussed other machine learning and deep learning techniques including references and discussing the advantages and disadvantages of their use.

Line 41-42: Instead of using the phrase 'a new paradigm,' it would be more accurate to reference earlier studies identifying the connection between ENSO and other tropical basins. For example, the connection between ENSO and the Indian Ocean Dipole (IOD) was first identified by Saji et al. (1999): A Dipole Mode in the Tropical Indian Ocean (Nature). This study demonstrated how the IOD can influence ENSO dynamics and has been foundational in the field.

Thanks for the comments. As we use Spy4CAST to predict ENSO using the Equatorial Atlantic as predictor, we mentioned just this paradigm. Nevertheless, we have considered convenient to also include other tropical basins impacting ENSO, as the one suggested by the referee. Our methodology could also be used and, in fact, following Saji et al (1999) we are also going to include in the new version the prediction of ENSO from the Indian Ocean.

Line 46-47: reliable? The citation at the end of the sentence is incomplete.

Thanks for the comment. We want to apologize as we did not include the complete reference, which is now included and it is Counillon et al. [2021].

Line 56: '... not designed to assess stationarity' seems to contradict Line 250: 'Spy4Cast is able to perform a validation methodology to look for non-stationary relations.' Line 260 as well.

We have corrected this sentence and also provided an example of an assessment of stationarity, as in Rodríguez-Fonseca et al. [2016].

**Line 59: fix the reference**

Thank you for the comment, we will fix it.

Line 69: unit tests?

Thank you for the comment. The spelling is fixed now.

Line 76: Section 4 is an example of using Atlantic to predict Pacific SSt. However, if you have the Sahelian rainfall example, I think it would be useful to include in your manual/documentation and showcase different settings and functionalities of your work.

Thank you for pointing that out. All the examples are included in the documentation and in the manual. Also, in the new version of the manuscript, the snippets of code have been removed to improve readability.

**Line 149: only being able to take monthly data seems limited capability to me.**

The model could run with daily data for other applications, but it is specifically designed for seasonal predictability, in which monthly to seasonal means are used. Nevertheless, it is not difficult to change the Preprocess class to do it.

Line 190: How do you determine the sample size? Monthly data are likely highly correlated, i.e., each month is not an independent point, you need to use the effective sample size when you do statistical analysis.

**We are not using consecutive monthly data. The sample interval is normally 1 year.**

Line 195, 197 and more: What table or listing are you referring to? In general, when referencing your previous paper on which this software is built, I suggest specifying the relevant tables, listings, or figures. This would make it easier for users to trace the code development and better understand the overall concept.

Thank you for your comment. In those lines we refer to the documentation and to the manual. We are aware it is not easy to follow reference this way so in the newer version we have made explicit reference to the documentation and the manual instead of the previous citations.

**Line 223: It says 2010 in the listing.**

Thank you for pointing it out. we have removed the listings in the revised manuscript.

Line 224: You mention non-stationarity multiple times in the manuscript, but it is unclear how you determine it. Since there are various methods to assess non-stationarity, I recommend specifying the approach you used to ensure clarity.

Thanks for the comment. We think that the text was confusing because one can validate a model for a period, training the model with the data of another period, and find skillful prediction because the signal is stationary. But also, it can happen that, training the model with the data of another period we don't find skillful predictions because the signal is not stationary. So, this is not the way to test stationarity a priori. So, we have re-writen the section to better explain the usefulness of the validation and a particular application for non stationary signals. In addition, in this new version of the manuscript we have better specified the way we can assess non-stationarity. There is a parameter associated with each of the modes of variability, ruv, which gives the degree of linear relationship between the expansion coefficients of a particular mode. If this number is not close to 1, that means that the relationship is not robust along the whole period. One of the most common techniques to assess this non-stationary behavior is to calculate moving correlations between the expansion coefficients. In the Github we have included an example about the relation between the anomalous boreal spring rainfall in Europe and Pacific SSTs as the one shown in Rodríguez-Fonseca et al. [2016] to illustrate how can we use Spy4cast to assess non-stationarity: Pacific\_Impact\_European\_Rainfall.ipynb

**Line 225: What does 'a hot spot' in climate variability studies mean?**

Sorry for the error. We meant "hot topic". This is now corrected

**Line 241: I think 'can be represented' is a more accurate phrasing.**

Thank you, we have rephrased it taking it into account.

**Line 246 Us should be in math form?**

Thank you for pointing it out.

Line 249: The rest of modes... this statement is misleading and not accurate. Fig 5 seems to say 68% instead of 76%?

Thanks for the comment. We have changed this statement to address this comment.

**Line 254: can you say what years? 94 and 91 for example?**

Thanks for the comment. It is true that not all the years, for example 1994 and 1995, present skill. We were talking about those years in which the modes presents high values in the expansion coefficients, indicating years in which there was an Atlantic Niño(Niña) and a Pacific La Niña (El Niño). We have rewritten the sentence to better indicate the years we refer to.

**Line 260: I am not sure you have explained how to use your software to determine stationarity**

Thanks for the comment. It is true that we have not indicated how to asses stationarity. In this new version we have added an example in the Github and in the manual of the API in which the stationarity is assessed. It is indicated in a previous comment.

**Line 265: This is not a 'new' approach but rather a ready-to-use software implementation of a well-established method for seasonal forecasting.**

Thanks for the comment. It is true that the methodology is well established but the python approach not. We have to admit that the sentence was pretencious and we have changed it, indicating, as suggested by the referee that "Spt4CAST is a ready-to-use software implementation of a the MCA methodology to be used to asses seasonal forecast but also non stationarities"

**Line 270: What is OFF project again?**

In the new version of the manuscript we have indicated what is this project. "OFF: Ocean For Future" is a National Project TED2021-130106B-I00) from the European Commission "Next Generation" funds in which predictability is assessed for all CMIP6 model under different scenarios using this tool coupled to the ESMVALTOOL. We have acknowledged this project in this manuscript.

**Figure 3 caption: You need to label what year this is. Is it 1997 based on List 4?**

Thank you for comment. We have changed the caption to make it clear.

**Listing 6: Do you need to 'import Preprocess' first in this script?**

Yes, it was a typo but, as it has been explained previously, listings have been removed in the new version of the manuscript.

**References**

- François Counillon, Noel Keenlyside, Thomas Toniazzo, Shunya Koseki, Teferi Demissie, Ingo Bethke, and Yiguo Wang. Relating model bias and prediction skill in the equatorial atlantic. *Climate Dynamics*, 56:2617–2630, 2021.
- Ibrahima Diouf, Roberto Suárez-Moreno, Belen Rodríguez-Fonseca, Cyril Caminade, Malick Wade, Wassila M Thiaw, Abdoulaye Deme, Andrew P Morse, Jaques-André Ndione, Amadou T Gaye, et al. Oceanic influence on seasonal malaria incidence in west africa. Weather, Climate, and Society, 14(1): 287–302, 2022.
- James B Elsner and CP Schmertmann. Assessing forecast skill through cross validation. Weather and Forecasting, 9(4):619–624, 1994.
- Åke Johansson, Anthony Barnston, Suranjana Saha, and Huug Van den Dool. On the level and origin of seasonal forecast skill in northern europe. *Journal of the Atmospheric Sciences*, 55(1):103–127, 1998.
- Armin Köhl and Andrey Vlasenko. Seasonal prediction of northern european winter air temperatures from sst anomalies based on sensitivity estimates. *Geophysical Research Letters*, 46(11):6109–6117, 2019.

- Jorge Lopez-Parages, Iñigo Gómara, Belén Rodríguez-Fonseca, and Jesús García-Lafuente. Potential sst drivers for chlorophyll-a variability in the alboran sea: A source for seasonal predictability? *Frontiers* in Marine Science, 9:931832, 2022.
- Joel Michaelsen. Cross-validation in statistical climate forecast models. Journal of Applied Meteorology and Climatology, 26(11):1589–1600, 1987.
- Belén Rodríguez-Fonseca, Roberto Suárez-Moreno, Blanca Ayarzagüena, Jorge López-Parages, Iñigo Gómara, Julián Villamayor, Elsa Mohino, Teresa Losada, and Antonio Castaño-Tierno. A review of enso influence on the north atlantic. a non-stationary signal. *Atmosphere*, 7(7):87, 2016.
- Eroteida Sánchez García, José Voces Aboy, and Ernesto Rodríguez Camino. Calibration and combination of seasonal forecast over southern europe. 2014.
- P Sinha, UC Mohanty, SC Kar, SK Dash, AW Robertson, and MK Tippett. Seasonal prediction of the indian summer monsoon rainfall using canonical correlation analysis of the ncmrwf global model products. *International journal of climatology*, 33(7), 2013.
- Koen Verbist, Andrew W Robertson, Wim M Cornelis, and Donald Gabriels. Seasonal predictability of daily rainfall characteristics in central northern chile for dry-land management. *Journal of Applied Meteorology and Climatology*, 49(9):1938–1955, 2010.
- Daniel S Wilks. Probabilistic canonical correlation analysis forecasts, with application to tropical pacific sea-surface temperatures. *International journal of climatology*, 34(5), 2014.
- DS Wilks. Northern hemisphere snow cover, indo-pacific ssts, and recent trend as statistical predictors of seasonal north american temperature. *Journal of Applied Meteorology and Climatology*, 54(1):58–68, 2015.
- Eleni Xoplaki et al. *Climate variability over the Mediterranean*. PhD thesis, Verlag nicht ermittelbar, 2002.